# Exosomes define a local and systemic communication network in healthy pancreas and pancreatic ductal adenocarcinoma

Bárbara Adem[1,2], Nuno Bastos[1,2], Carolina F. Ruivo[1], Sara Sousa-Alves [1], Carolina Dias [1,3], Patrícia F. Vieira [1,3], Inês A. Batista [1,2], Bruno Cavadas[1], Dieter Saur [4,5], José C. Machado[1,6,7], Dawen Cai [8,9,10] & Sonia A. Melo [1,6,7] ✉

Pancreatic ductal adenocarcinoma (PDAC), a lethal disease, requires a grasp of its biology for effective therapies. Exosomes, implicated in cancer, are poorly understood in living systems. Here we use the genetically engineered mouse model (ExoBow) to map the spatiotemporal distribution of exosomes from healthy and PDAC pancreas in vivo to determine their biological significance. We show that, within the PDAC microenvironment, cancer cells establish preferential communication routes through exosomes with cancer associated fibroblasts and endothelial cells. The latter being a conserved event in the healthy pancreas. Inhibiting exosomes secretion in both scenarios enhances angiogenesis, underscoring their contribution to vascularization and to cancer. Inter-organ communication is significantly increased in PDAC with specific organs as most frequent targets of exosomes communication occurring in health with the thymus, bone-marrow, brain, and intestines, and in PDAC with the kidneys, lungs and thymus. In sum, we find that exosomes mediate an organized intra- and inter- pancreas communication network with modulatory effects in vivo.

Pancreatic ductal adenocarcinoma (PDAC) is the prevailing form of pancreatic cancer, characterized by its advanced stage and limited treatment options, leading to a grim prognosis with an average survival time of only 6 months after diagnosis[1,2]. Despite extensive efforts, our current understanding of PDAC biology has not translated into significant improvements in patients' outcome. Approaches and strategies employed thus far to identify potential therapeutic avenues have yielded limited breakthroughs, leaving a dire need for novel insights into the intricate biology of PDAC to improve patient care.

Intercellular communication plays a pivotal role in maintaining homeostasis and orchestrating disease processes. Among the key mediators of intercellular communication are extracellular vesicles (EVs), which are believed to be released by all cell types[3]. EVs encompass two main subclasses, namely exosomes and microvesicles[4]. Exosomes, nanosized vesicles ranging from 30–150 nm, originate from the endocytic pathway, while microvesicles directly bud from the plasma membrane and can reach several micrometers in size[5]. Exosomes have emerged as major contributors in various biological processes that promote tumor progression[6]. These vesicles are enriched with tetraspanins such as CD63, CD81, and CD9, and are released by exocytosis events that follow the docking and fusion of multivesicular bodies with the plasma membrane, a process mediated by Rab27a[7].

[1]i3S—Instituto de Investigação e Inovação em Saúde, Universidade do Porto, Porto, Portugal. [2]Instituto de Ciências Biomédicas de Abel Salazar, Universidade do Porto, Porto, Portugal. [3]Faculdade de Medicina, Universidade do Porto, Porto, Portugal. [4]Medical Clinic and Polyclinic II, Klinikum rechts der Isar, Technical University Munich, Munich, Germany. [5]German Cancer Research Center (DKFZ) and German Cancer Consortium (DKTK), Heidelberg, Germany. [6]Departamento de Patologia, Faculdade de Medicina, Universidade do Porto, Porto, Portugal. [7]P.CCC Porto Comprehensive Cancer Center Raquel Seruca, Porto, Portugal. [8]Department of Cell and Developmental Biology, Medical School, University of Michigan, Ann Arbor, MI, USA. [9]Biophysics, LS&A, University of Michigan, Ann Arbor, MI, USA. [10]Michigan Neuroscience Institute, University of Michigan, Ann Arbor, MI, USA. ✉e-mail: smelo@i3s.up.pt

Therefore, Rab27a is indispensable for the secretion of exosomes and vesicles carrying endosomal markers like CD63[7,8].

Exosomes exert their influence on target cells through their cargo, which encompasses proteins, DNA, RNA, lipids, and metabolites, delivered via endocytosis or direct fusion with the recipient cell's plasma membrane[9–11]. Additionally, exosomes can engage in receptor-ligand interactions to modulate target cells[9]. It is hypothesized that the impact of exosomes spans a wide range of processes, including immune response modulation, tumor microenvironment (TME) remodeling, angiogenesis, migration and invasion of cancer cells, and the establishment of pre-metastatic niches[6,12]. However, the communication routes established by endogenously produced exosomes remain largely unexplored. Direct evidence of local as well as inter-organ communication mediated by exosomes remains an open question, leaving largely uncharted the biological significance of exosomes in a multicellular organism. Thus, our aim is to unveil the intra-pancreas and inter-organ spatiotemporal distribution patterns of exosomes in PDAC and healthy pancreas to uncover its biological significance and potential therapeutic targets.

To accomplish this aim, we have developed an exosomes tagging reporter system, the ExoBow, enabling tissue- and cell-type-specific tracking of CD63 positive exosomes (CD63+ Exos) in vivo. This study comprehensively dissects the spatiotemporal distribution of naturally secreted pancreas exosomes locally (intra-pancreas) and systemically (inter-organ). Combining the ExoBow mouse with well-established PDAC models, we identify the routes of communication established by the healthy pancreas and the PDAC cells.

## Results

### The ExoBow model to lineage-trace exosomes

To identify the spatiotemporal distribution of pancreas exosomes and determine their biological significance, we developed a genetically engineered mouse model (GEMM), ExoBow, that enables tracing of naturally produced exosomes. Our strategy involved tagging the exosomes marker CD63[8,13,14], which we confirmed to be present in distinct human PDAC-derived exosomes (Supplementary Fig. 1a), with various fluorescent proteins. To determine whether expression of tagged CD63 would affect tumor growth kinetics, we established a human PDAC cell line stably expressing CD63-GFP and orthotopically implanted the clone and its parental counterpart in the pancreas of immunodeficient mice (Supplementary Fig. 1b, c). No significant differences in tumor growth and weight were observed during disease progression and at time of euthanasia (Supplementary Fig. 1d, e).

The ExoBow transgene ($R26^{CD63-XFP/+}$) is a dual recombinase-driven model (Cre and flippase−Flp) that can be conditional to a target organ or cell-type (Fig. 1a). The transgene cassette was designed to be inserted into intron 1 of the *ROSA26* locus and is controlled by the *CAG* promoter. A STOP cassette flanked by Frt sites was placed upstream of the CD63 open reading frame to prevent its expression. The mouse *CD63* sequence is followed by four different fluorescent proteins: mCherry, phiYFP, eGFP and mTFP, which are flanked by distinct and incompatible lox recombination sites (*LoxN*, *Lox2272* and *Lox5171*), similar to the Brainbow2 design[15,16]. Upon Flp recombination, removal of the STOP cassette leads to the expression of CD63-mCherry fusion protein (CD63-mCherry+) that marks secreted exosomes. Additional Cre recombination in the same cell leads to the removal of mCherry and the expression of either CD63-phiYFP, CD63-eGFP, or CD63-mTFP fusion protein (CD63-XFP+). We evaluated the successful recombination of the ExoBow transgene in mouse embryonic stem cells (ESCs) (Supplementary Fig. 1f, g). Here, we focused on the study of the heterozygous model (Supplementary Fig. 1h).

To investigate the intercellular localization of CD63-XFP proteins we cloned each mouse CD63-XFP protein from the ExoBow transgene into the pRP[Exp]-Puro-CAG backbone vector. These constructs were then separately transfected into a human PDAC cell line (BxPC-3;

Supplementary Fig. 2a). Stable clones were established and the expression of the fusion protein confirmed by flow cytometry (Supplementary Fig. 2b). Our results revealed that the expression of the CD63-XFP proteins mirrored that of the endogenous (human) CD63 protein, displaying a speckle-like pattern with accumulation near the nuclei, indicative of its expected endosomal localization (Supplementary Fig. 2c, d)[17]. Using antibodies specific to each fluorescent protein and to human CD63, we further confirmed co-localization of the proteins, providing evidence that the fusion of CD63 with fluorescent proteins does not disrupt its cellular localization (Supplementary Fig. 2e). We further validated the pattern of CD63-XFP expression in a panel of cell lines established from PDAC ExoBow GEMMs (Supplementary Fig. 3).

To confirm that CD63-XFP expression does not interfere with exosomes secretion, we quantified the total number of vesicles released per cell by nanoparticle tracking analysis. No significant differences between the CD63-XFP clones and the parental cell line were observed (Supplementary Fig. 4a). Exosomes were isolated by ultra-centrifugation and the expression of the CD63-XFP in each clone was confirmed by western-blot (Supplementary Fig. 4b). Further characterization involved exosomes isolation by ultracentrifugation followed by continuous sucrose gradient fractionation. Protein samples from fractions corresponding to distinct densities were then subjected to western-blot analysis using anti-XFP antibodies. This analysis demonstrated that the various CD63-XFP clones secrete color-coded exosomes (Supplementary Fig. 4c). Imaging flow cytometry analysis of exosomes revealed the enrichment in the secreted exosomes that are CD63-XFP positive for each BxPC-3 clone (Supplementary Fig. 4d). In addition, exosomes were isolated by ultracentrifugation followed by size exclusion chromatography. We confirmed that the CD63-XFP positive fractions corresponded to the fractions positive for exosomes markers such as syntenin or CD63, and negative for cytochrome C or apolipoprotein A1 (Supplementary Fig. 4e, f).

In summary, we designed a Flp or Flp/Cre dependent ExoBow reporter that generates tagged CD63+ exosomes without interfering with CD63 protein cellular localization nor with exosomes secretion.

### Targeting the ExoBow to healthy pancreas and PDAC

In vivo imaging shows that crossing the ExoBow model to an Flp driver line under the control of the Pdx1 pancreas-specific promoter (*Pdx1-Flp*; $R26^{CD63-XFP/+}$ hereafter referred to as Panc-CD63-mCherry[18,19]), results in strong mCherry fluorescence in the pancreas (Fig. 1b). Immunofluorescence analysis further confirmed efficient recombination and CD63-mCherry expression in pancreas cells (Fig. 1c), which was also validated by flow cytometry (Supplementary Fig. 5a). To confirm the presence of CD63-mCherry in pancreas-derived exosomes, we fractionated EVs from the pancreas of the Panc-CD63-mCherry mouse model into small and large EVs. We found CD63-mCherry to be enriched in the small EVs fraction, which indicates an enrichment in the exosomes subpopulation (Fig. 1d).

Crossing the *Pdx1-Flp*; $R26^{CD63-XFP/+}$ model to an additional *Pdx1-Cre* driver line allows the stochastic fusion of eGFP, phiYFP or mTFP to CD63 (CD63-XFP) in individual pancreatic cells (Panc-ExoBow, Fig. 1e). Close investigation by high power microscopy indicates that CD63-XFP localizes in small vesicles and plasmid membrane structures in the cell (Fig. 1f). By immunofluorescence we could observe co-localization with Alix, syntenin and Rab7 endosomal markers. However, Rab7 co-localizes in a less extent since it can be associated with MVB directed to lysosomal compartments[20] (Supplementary Fig. 5b).

We observed clustering of pancreatic cells in which all cells express the same fluorescent protein fusion (Fig. 1f), supporting a common origin in recombined stem/progenitor cells as expected. All CD63-XFP fusion proteins were detected in vesicles isolated from the pancreas, once again with an enrichment in the smaller EVs fraction, characteristic of the exosomes subset (Fig. 1g). Further characterization of these fractions revealed that small EVs of *wild-type* (WT) mice

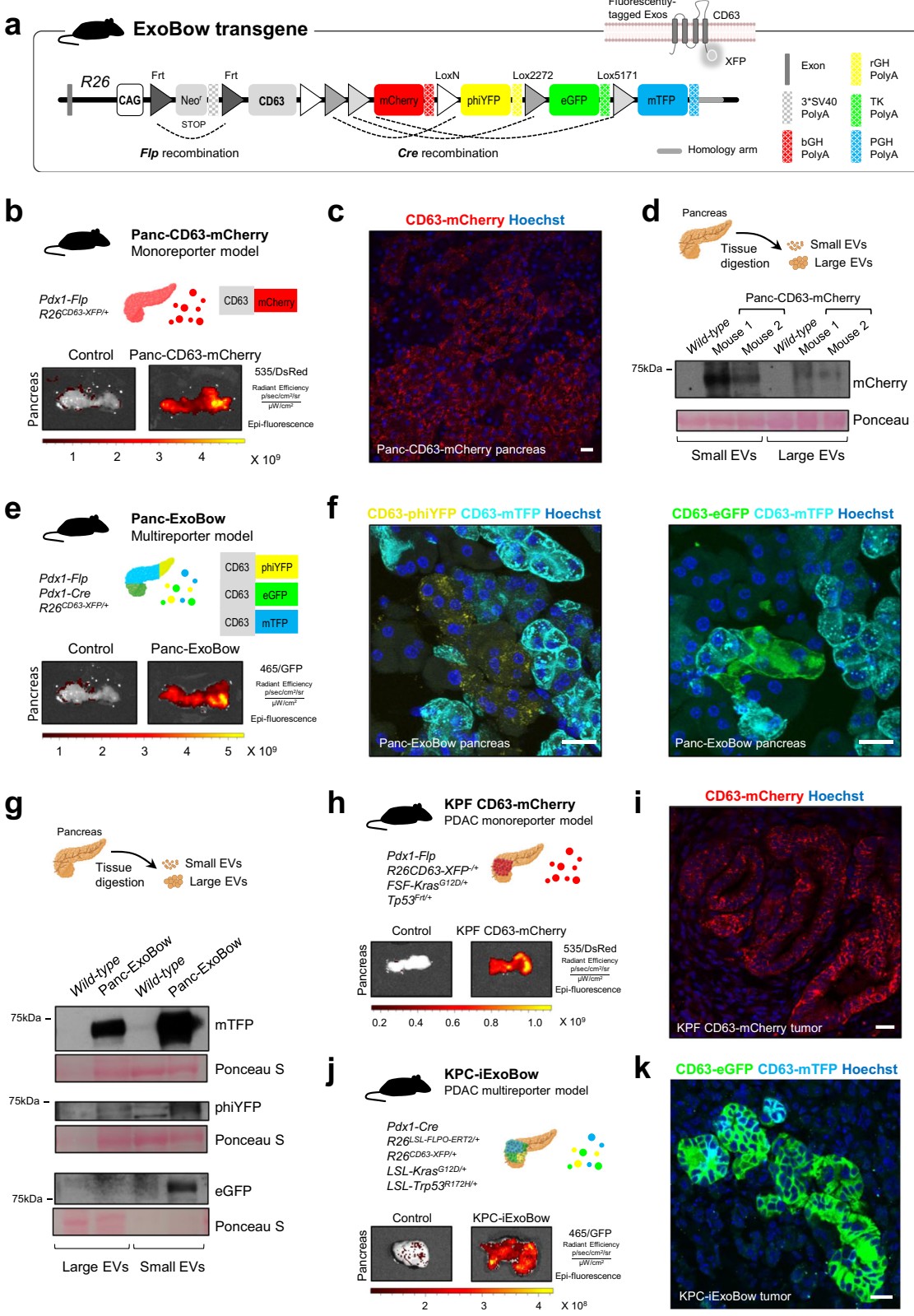

are enriched in CD63 in comparison to large EVs (Supplementary Fig. 5c). In addition, small EVs are positive for exosomes markers such as syntenin and Alix and negative for cytochrome C or apolipoprotein A1. Collectively, we can conclude that the CD63-XFP EVs fall in the exosomes category which is enriched in the small EVs fraction (Supplementary Fig. 5d).

To be noted, we still observed various levels of CD63-mCherry expression in the pancreas of Panc-ExoBow mice, which reflects incomplete Cre recombination (Supplementary Fig. 6a−c). To compare the Flp and Cre recombination efficiency, we performed PCR of the recombined ExoBow allele in both Panc-CD63-mCherry and Panc-ExoBow transgenic models (Supplementary Fig. 6d).

**Fig. 1 | The ExoBow transgene efficiently labels pancreas cells and its derived exosomes. a** The ExoBow construct is inserted in the intron one of *ROSA 26* (*R26*) locus and is under the action of a strong synthetic promotor (CAG). Upstream of the exosomal marker CD63 mouse gene there is a neomycin resistance cassette with a stop-codon flanked by Frt sites that prevents further transcription. Following CD63 there are 4 fluorescent reporters: mCherry, phiYFP, eGFP and mTFP, each with a polyA sequence. **b** Schematic of the monoreporter mouse model Panc-CD63-mCherry in which *Pdx1* drives the expression of the Flp recombinase. Pancreas imaging using IVIS Lumina System illustrating CD63-mCherry expression (535 excitation laser and DsRed emission filter). Pancreas of control (R26$^{CD63-XFP/+}$, no recombinases, left) and Panc-CD63-mCherry mice (right; experiment repeated with a total of 6 mice). **c** Confocal microscopy images of a maximum projection of a Panc-CD63-mCherry pancreas section depicting exocrine and endocrine CD63-mCherry positive cells. Immunofluorescence against mCherry (red; experiment repeated with a total of 3 mice). **d** Schematic representation of the isolation of interstitial EVs from the pancreas tissue according to Crescitelli et al.[53]. Anti-mCherry western-blot in small and large EVs fractions isolated from pancreas tissue of *wild-type* (WT, control) or Panc-CD63-mCherry mice (experiment repeated with a total of 3 mice). **e** Schematic representation of the multireporter mouse model, Panc-ExoBow, in which both Flp and Cre recombinases are under the control of *Pdx1* promoter. Pancreas imaging using IVIS Lumina System illustrating CD63-eGFP, CD63-phiYFP, and CD63-mTFP (465 excitation laser and GFP emission filter). Pancreas of control (R26$^{CD63-XFP/+}$, left) and Panc-CD63-mCherry mice (right; experiment

repeated with a total of 5 mice). **f** Confocal images of a maximum projection of a Panc-ExoBow pancreas section depicting CD63-mTFP, CD63-phiYFP and CD63-eGFP positive cells. Immunofluorescence for mTFP (cyan), phiYFP (yellow) and eGFP (green; experiment repeated with a total of 3 mice). **g** Schematic representation of the isolation of interstitial EVs from the pancreas tissue according to Crescitelli et al.[53]. Anti-mTFP, anti-phiYFP and anti-eGFP western-blot in small and large EVs fractions isolated from pancreas tissue of WT or Panc-ExoBow mice (experiment repeated with a total of 2 mice). **h** Schematic representation of the PDAC monoreporter mouse model, KPF CD63-mCherry. Pancreas images of control (no ExoBow transgene, left) and KPF CD63-mCherry mice (right; experimented repeated with a total of 4 mice). **i** Confocal microscopy images depicting CD63-mCherry positive cancer cells in the pancreas of a KPF CD63-mCherry mouse. Immunofluorescence against mCherry (red; experiment repeated with a total of 3 mice). **j** Schematic representation of the PDAC multireporter mouse model, KPC-ExoBow. Pancreas images of control (no ExoBow transgene, left) and KPC-ExoBow mice (right; experiment repeated with a total of 4 mice). **k** Confocal microscopy images depicting CD63-mTFP and CD63-eGFP positive cells in the pancreas of a KPC-ExoBow mouse. Immunofluorescence against mTFP (cyan), and eGFP (green; experiment repeated with a total of 2 mice). Mice age in **b**, **c**, **e**, **f** is of 8 weeks, **d** and **g** between 8–1 weeks, **h** i 16.3 weeks, and **j** k 17 weeks. In western-blots Ponceau S as loading control and 25 μg of protein samples was used. In all images nuclei are counterstained with hoechst (blue) and scale bars are 20 μm. Source data are provided as a Source Data file. Schemes created with BioRender.com.

Importantly, the expression of CD63-XFP in either transgenic model did not affect mouse development, pancreas function, or histology, neither the number of serum exosomes was altered (Supplementary Fig. 7a–e). In addition, similar size and number of both small and large pancreas derived vesicles were detected in both monoreporter and multireporter models compared to WT (Supplementary Fig. 7f, g).

To investigate the distribution of exosomes in PDAC, we crossed the ExoBow with two PDAC GEMMs that faithfully recapitulate the clinical and histopathological features of human PDAC[18,19,21]. By crossing the Flp-driven KPF with the ExoBow allele, termed KPF-CD63-mCherry, we achieved Flp-mediated expression of CD63-mCherry specifically in cancer cells (*Pdx1-Flp; R26$^{CD63-XFP/+}$; FSF-Kras$^{G12D/+}$; Trp53$^{Frt/+}$*; Fig. 1h,i). To induce CD63-XFP expression, we crossed the Cre-driven KPC model with ExoBow to generate the KPC-ExoBow Flp negative (*LSL-Kras$^{G12D/+}$; LSL-Trp53$^{R172H/+}$; Pdx1-Cre; R26$^{CD63-XFP/+}$*), which requires an additional Flp allele to promote CD63-XFP stochastic expression in PDAC. To demonstrate the versatility and applicability of the KPC-ExoBow transgene, we used three different Flp-based approaches. In the first approach, we injected adenovirus to express Flpo (Ad-CMV-Flpo) orthotopically in the pancreas of 6 weeks old KPC-ExoBow Flp negative mice (Supplementary Fig. 8a). In the second approach, we included an additional *Pdx1-Flp* allele, which allows recombination of the ExoBow transgene specifically in pancreas cells when *Pdx1* promoter is first active[22]. In the third approach, we included an additional *R26$^{LSL-FLPoERT2/+}$* allele to create the KPC-iExoBow model, which allows tamoxifen-inducible Flp recombination in PDAC[23] (Fig. 1j, k). No major differences were observed between the different reporter models, hereinafter all referred to as the KPC-ExoBow, concerning pancreas tumor histology or disease progression (Supplementary Fig. 8b).

In summary, we show that the ExoBow is a versatile transgenic model that can efficiently label secreted CD63$^+$ exosomes in pancreatic cells in the healthy pancreas or in PDAC without disruption of normal pancreas development or disease progression.

### Exosomes-mediated intra-pancreas communication is defined by specific routes that target endothelial cells and cancer-associated fibroblasts

The PDAC microenvironment exhibits a robust desmoplastic reaction, characterized by a multitude of cells that outnumber cancer cells[24,25].

We show that cancer associated fibroblasts (CAFs), endothelial and immune cells received PDAC CD63$^+$ Exos in the tumors of KPF CD63-mCherry and KPC-ExoBow mice (Fig. 2a). Communication with CAFs (CD140A$^+$ and α-smooth muscle actin-positive, αSMA$^+$) and endothelial cells (CD31$^+$; Fig. 2a,b) was more frequent than with cells of the immune system (CD45$^+$), as determined by the percentage of cells positive for CD63$^+$ Exos. Interestingly, rates of communication, defined by the number of cells positive for CD63$^+$ Exos with respect to their prevalence in the tumor, are not dependent on their frequency. We showed that the pattern of communication in PDAC is not a direct function of the number of cell subtypes (Fig. 2a and Supplementary Fig. 9a, b). Considering the prognostic significance of αSMA$^+$ CAFs in PDAC progression, we went to investigate how communication with these cells could impact their number or spatial distribution within the TME[26,27]. We first observed that PDAC lesions in KPC tumors present a heterogenous pattern of Rab27a expression, a surrogate marker of exosomes secretion (Fig. 2c). Next, we demonstrated that PDAC lesions that efficiently secrete exosomes (Rab27a$^{High}$) were surrounded by αSMA$^{Low}$ CAFs, typically associated with an inflammatory phenotype, while PDAC lesions that are Rab27a$^{Low}$ were surrounded by αSMA$^{High}$ CAFs[28] (Fig. 2d). This inverse correlation suggests that exosomes could modulate CAFs into an inflammatory phenotype and impact their spatial distribution within PDAC tumors.

Communication with endothelial cells occurs in a similar fashion in PDAC as well as in healthy pancreas, and the frequency of endothelial cells does not change upon malignant transformation (Fig. 3a–c and Supplementary Fig. 9c). What is more, communication with endothelial cells is the most prevalent route of communication, just after CAFs (Fig. 2a). Hence, we investigated how angiogenesis is modulated by exosomes in the healthy pancreas and in PDAC. We blocked the secretion of exosomes by conditionally knocking out (KO) Rab27a in the healthy pancreas (*Pdx1-Flp; Rab27a$^{Frt/Frt}$*) and in PDAC cancer cells using the fast progression PDAC model (PKT iRab27a, *Ptf1a-Cre; LSL-Kras$^{G12D/+}$; Tgfbr2$^{loxP/loxP}$; R26$^{LSL-FLPoERT2/+}$; Rab27a$^{Frt/Frt}$*). The number of vessels, quantified by the number of CD31$^+$ endothelial cells, was significantly increased in Rab27a KO mice in both healthy pancreas and in PDAC (Fig. 3d–g), showcasing the capacity of pancreas and PDAC-derived exosomes to modulate angiogenesis.

It was previously suggested that cancer exosomes promote both an anti-tumor response and an immunosuppressive microenvironment[29]. Interestingly, we did not observe an increase in

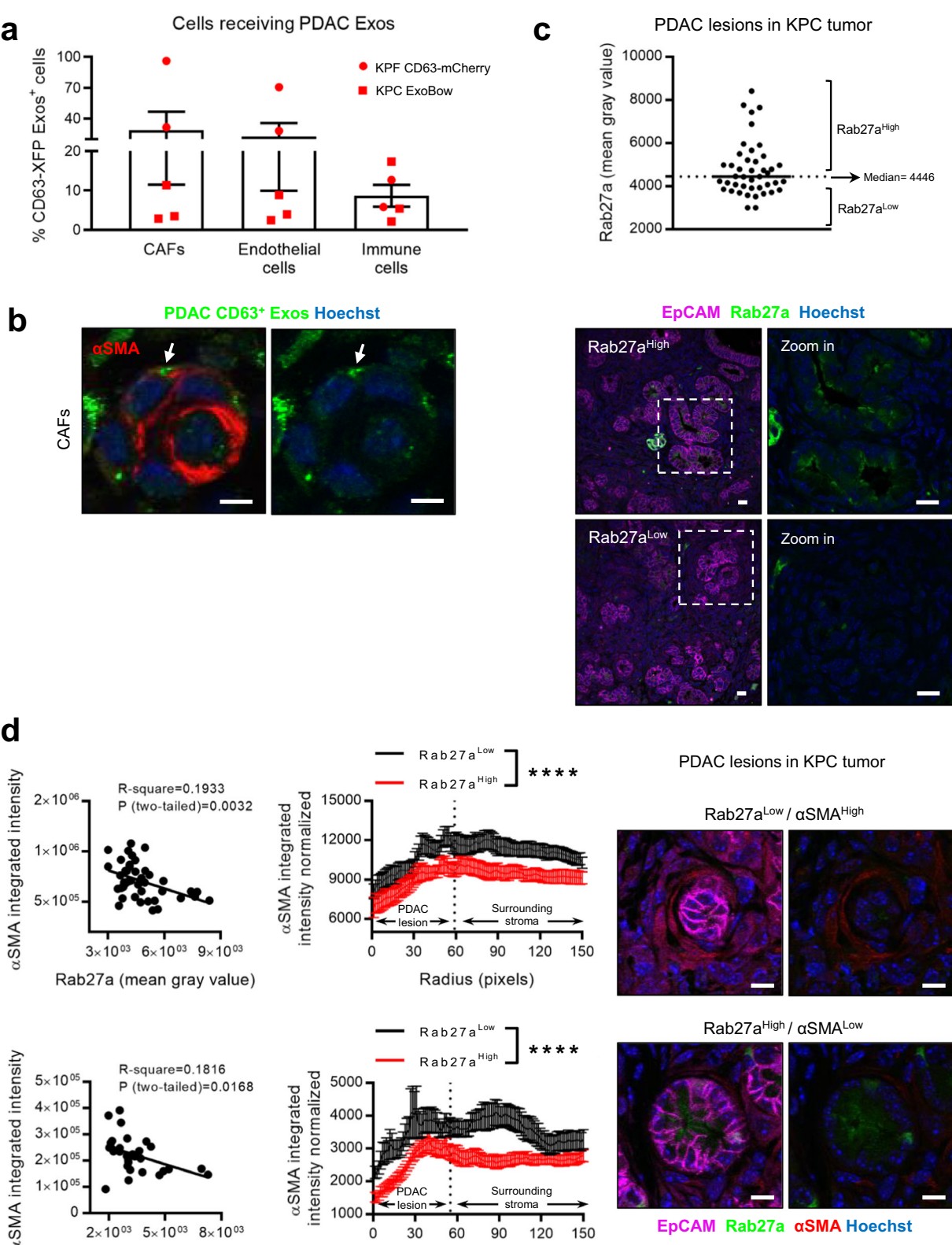

the percentage of immune cells that received CD63+ Exos, despite the significant increase in the percentage of immune cells population from healthy to PDAC (Fig. 4a, b). We found that PDAC CD63+ Exos are taken up by T cells (TCRβ+), cells of the monocyte lineage (CD11b+Ly6G/C−) and natural killer (NK) cells (CD11b+NK+) in increasing frequencies, and this communication is not significantly different from that occurring in the healthy pancreas (Fig. 4c and Supplementary Figs. 9d, e and 10). In

PDAC, NK cells are the ones that uptake PDAC CD63+ Exos at higher frequencies in comparison to the healthy pancreas (Supplementary Fig. 9d, e). In the healthy pancreas, cells of the monocyte lineage registered the highest levels of CD63+ Exos, and there was a decrease in this communication route upon malignant transformation. Here, we cannot disregard the phagocytic capacity of these cells, which could be taking up cell debris or non-exosomes vesicles. Finally, the lowest

**Fig. 2 | Exosomes mediate intra-pancreas communication in PDAC and CAFs spatial distribution. a** Dot plot representing the percentage of cancer-associated fibroblasts (CAFs; CD140A[+]), endothelial cells (CD31[+]) and immune cells (CD45[+]) that received PDAC CD63[+] Exos in tumors PDAC reporter mice analyzed by flow cytometry (n = 5 biologically independent animals). **b** Representative confocal microscopy images of PDAC Exos (green) accumulation in CAFs (alpha-smooth muscle actin −αSMA−in red). Nuclei were counterstained with hoechst (blue). Scale bar 5 μm. Experiment repeated with a total of 3 mice. **c** Dot plot representing the fluorescence intensity of Rab27a in different lesions (n = 43) of a KPC tumor, with representative confocal microscopy images of regions with Rab27a low or Rab27a high PDAC lesions. EpCAM (cancer cells) in magenta, Rab27a in green and nuclei were counterstained with hoechst (blue). Scale bar 20 μm. Dashed line represents the median of Rab27a intensity. **d** Linear regression of αSMA and Rab27a per PDAC lesion (left) and circular radial profile of the αSMA fluorescence intensity over centered Rab27a high or low PDAC lesions in two KPC tumors (upper (n = 43 PDAC lesions) and lower (n = 31 PDAC lesions) left) with representative confocal microscopy images of a Rab27a[Low]/αSMA[High] and Rab27aHigh/αSMA[Low]. Rab27a in green, αSMA in red and EpCAM in magenta, nuclei were counterstained with hoechst (blue). Scale bar 10μm. Dashed lines represent the median radius fitted to the manual PDAC lesions' ROI. Kolmogorov-Smirnov, ****p < 0.0001. Data are Mean ± SEM. Source data are provided as a Source Data file.

levels of communication were detected with T cells in PDAC and the healthy pancreas, despite a tendency towards an increase in the frequency of these cells in PDAC (Supplementary Fig. 9d, e). Interestingly, within T cell subpopulations, the same levels of communication were observed between T helper (TCRβ[+]CD4[+]) or cytotoxic T cells (TCRβ[+]CD4[-]), but decreased levels with regulatory T cells (CD4[+]Foxp3[+]; Fig. 4d, e).

Collectively, our data demonstrates that communication in vivo occurs in specific routes and is not a random event that depends on the number of cells. This observation comes in line with our recent findings in which we unraveled an organized and hierarchical intra-tumor communication network in PDAC[30]. Our findings suggest that angiogenesis is one of the major processes modulated by exosomes in the healthy pancreas and PDAC, and that PDAC exosomes are involved in the spatial distribution of CAFs in the TME.

## Exosomes-mediated inter-organ communication is enhanced in PDAC and entails specific communication routes in cancer with thymus, kidneys and lungs

Comparing the overall levels of fluorescence across all organs, PDAC exhibited a significant increase in inter-organ communication mediated by exosomes compared to the healthy pancreas (Fig. 5a). Specifically, there was a 16-fold increase in the communication rate of CD63[+] Exos in PDAC, with 20% more organs testing positive for PDAC CD63[+] Exos in comparison to the healthy pancreas (Fig. 5b and Supplementary Fig. 11a). This increase is evenly distributed among the different organs with only axillary lymph nodes (AXL LN) and the mid-section of the intestine being higher in a healthy context (Supplementary Fig. 11b). By mapping the inter-organ distribution of CD63[+] Exos in healthy pancreas and in PDAC we showed that PDAC CD63[+] Exos consistently accumulate mostly in the kidneys, thymus, and lungs (Fig. 5c). Although relatively lower compared to PDAC, healthy pancreas CD63[+] Exos were also detected in the thymus, brain, femur/bone-marrow, and different sections of the intestines (Supplementary Fig. 11c, d). The stomach, proximal intestine, colon, salivary glands, brain (PDAC) and the liver (late PDAC) were excluded from the analysis due to known leakage of the *Pdx1* promoter[18], presence of the recombined ExoBow allele, and/or presence of histologically confirmed metastatic foci.

To analyze inter-organ communication at early stages of the disease, we utilized KPF CD63-mCherry and KPC-ExoBow mice (Fig. 5d and Supplementary Fig. 11e, f). Communication was significantly increased in an early PDAC context in comparison to the healthy pancreas, being even greater in late stages of PDAC (Fig. 5a, e). However, we did not observe differences in the target organs between early and late PDAC stages (Fig. 5d). Transitioning from a pre-malignant stage (healthy pancreas) to early PDAC, we observed increasing communication with the lungs, kidneys, and heart (Fig. 5d−f). Notably, communication rates with the kidneys consistently increased throughout disease progression, reaching peak levels in late PDAC stages, while communication with the thymus fluctuates with high communication in the healthy pancreas, and at late PDAC stages (Fig. 5d). These findings underscore the dynamic nature of

communication mediated by exosomes in vivo, highlighting that the distinct communication routes are dependent on specific biological contexts.

To determine the recipient cell types of CD63[+] Exos in the most frequent sites of communication in PDAC, the lung and the kidneys, we employed cell markers for prevalent cell populations in these organs. For the kidneys, we examined megalin-positive cells (proximal tubules), aquaporin-2-positive cells (collecting ducts), and podoplanin-positive cells (glomeruli). Our observations revealed that proximal tubules (Megalin[+]) were the primary recipients of PDAC CD63[+] Exos, followed by collecting ducts (Aquaporin-2[+]), in both early and late PDAC stages (Fig. 6). Interestingly, kidney glomeruli (Podoplanin[+]) did not show positivity for PDAC CD63[+] Exos in both scenarios. This immunofluorescence pattern aligned with our IVIS assessment, which demonstrated an increase in PDAC EVs accumulation at late stages compared to early disease stages (Fig. 5d, f and Fig. 6a, b).

In the lungs, we sought uteroglobin-positive cells (non-ciliated epithelial Clara cells), podoplanin-positive cells (type-I pneumocytes), and Transcription Termination Factor 1 (TTF1)-positive cells (type II alveolar cells and club cells). We found that Clara cells (Uteroglobin[+]) and type-I pneumocytes (podoplanin[+]) were the recipients of PDAC CD63[+] Exos, while no instances of communication were observed for type II alveolar cells and club cells (TTF1[+], Fig. 6a, b). These communication patterns remained consistent in both early and late PDAC stages in the lungs (Fig. 5d, f and Fig. 6a, b).

Moreover, we assessed the number of small EVs derived from healthy and PDAC pancreas, finding an enrichment in small EVs in the cancer context (Fig. 6c). This trend evident in the number of vesicles detected in the circulation in PDAC-bearing mice compared to healthy ones (Fig. 6d). Importantly, we detected the presence of PDAC CD63[+] Exos in the circulation of PDAC mice, with a greater prevalence at later PDAC stages (Fig. 6e). This observation corresponded with the increase in CD63[+] Exos biodistribution levels across different organs from early to late PDAC stages (Fig. 5d). Collectively, these results contribute for a comprehensive understanding of the long-distance intercellular communication network established by PDAC CD63[+] Exos and their potential biological significance. In summary, our study provides evidence of in vivo communication by exosomes with distant organs originating from both the healthy pancreas and PDAC. We demonstrated that inter-organ communication is enhanced in PDAC and is not a random event, following specific communication particularly to the thymus in the healthy pancreas, and the thymus, kidneys, and lungs in PDAC.

## Comprehensive analysis of PDAC small EVs cargo reflects the in vivo local inter-connectome

To gain insights into the mechanisms by which PDAC exosomes influence recipient cells upon uptake, we first isolated small EVs from both healthy WT pancreas and tumors from KPC mice and conducted mass spectrometry analysis (Fig. 7a). Our analysis revealed that these two small EVs populations shared a substantial portion of their cargo (71%, Fig. 7b), potentially indicative of common ancestry and

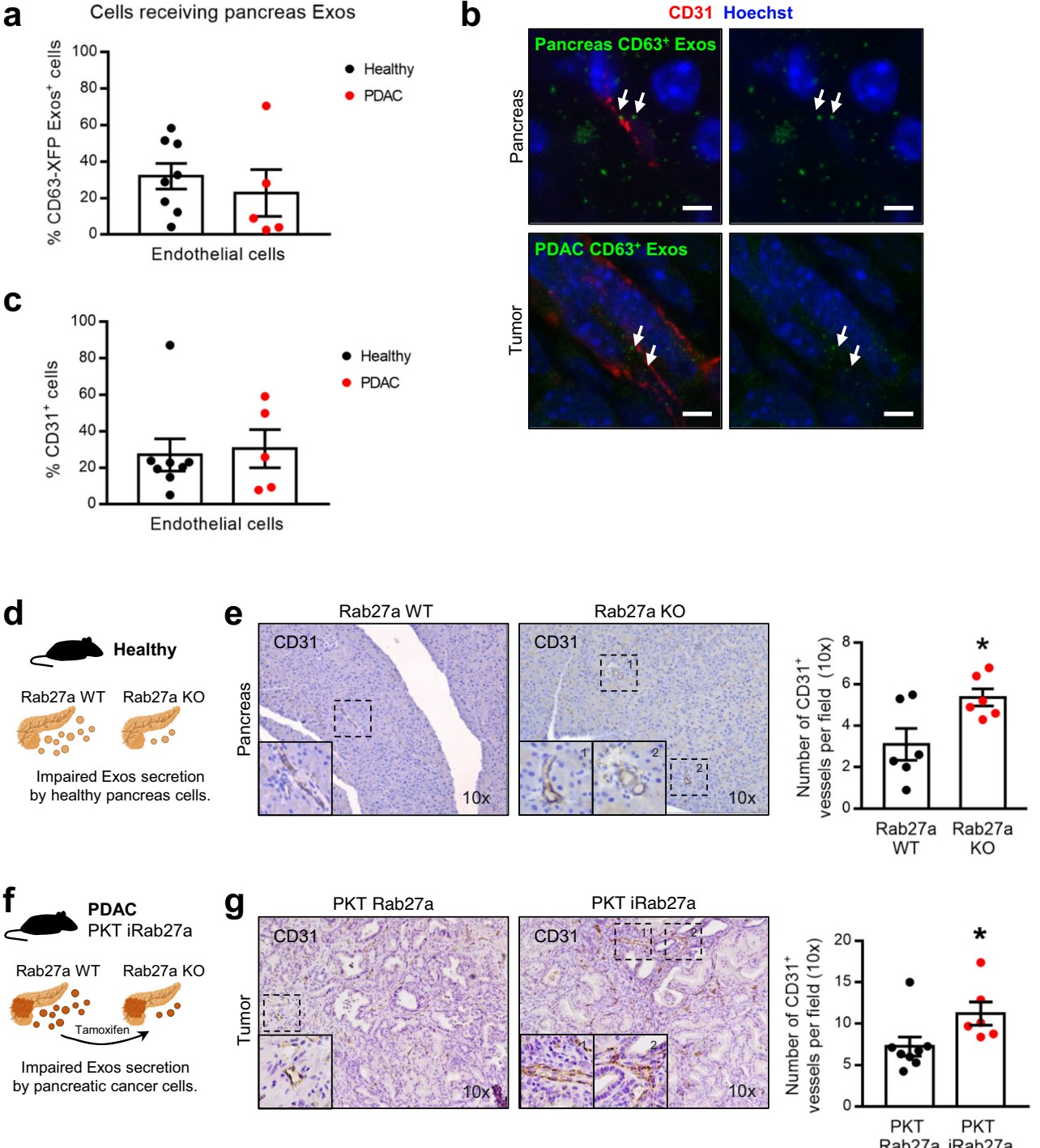

**Fig. 3 | Pancreas exosomes mediate local communication and constrain angiogenesis. a** Dot plot representing the percentage of endothelial cells (CD31+) that received pancreas-derived CD63+ Exos in health (Panc-CD63-mCherry, $n = 8$) and in PDAC (KPF CD63-mCherry $n = 2$ and KPC-ExoBow $n = 3$) analyzed by flow cytometry. **b** Representative confocal microscopy images of pancreas-derived Exos (green) accumulation in endothelial cells (CD31+ in red) in healthy pancreas of Panc-CD63-mCherry mice (upper panel) or tumors of KPF CD63-mCherry mice. Nuclei were counterstained with hoechst (blue). Scale bar 5 μm. **c** Dot plot representing the percentage of endothelial cells (CD31+) in the pancreas microenvironment in health (Panc-CD63-mCherry, $n = 8$) and in PDAC (KPF CD63-mCherry $n = 2$ and KPC-ExoBow $n = 3$) analyzed by flow cytometry. **d** Schematic representation of the healthy Rab27a KO GEMM in which pancreas cells have impaired secretion of

exosomes. **e** Representative CD31 IHC images (10 x, left) and respective quantification (right) in the pancreas of *wild-type* (Rab27aWT, $n = 6$) and Pdx1 Rab27aFrt/Frt (Rab27aKO, $n = 6$) mice. Two-tailed unpaired *t*-test, $p = 0.0261$. **f** Schematic representation of the PDAC Rab27a KO GEMM in which pancreas cells have impaired secretion of exosomes upon tamoxifen administration. The PKT Rab27a model is the control group which lacks the R26LSL-Flpo-ERT2 allele, hence upon tamoxifen administration expresses Rab27a and has proficient exosomes secretion. **g** Representative CD31 IHC images (10 x, left) and respective quantification (right) in the pancreas of control PKT Rab27a ($n = 8$) and PKT iRab27a ($n = 6$) mice. Two-tailed unpaired *t*-test, $p = 0.0485$. Data are Mean ± SEM. Source data are provided as a Source Data file. Schemes created with BioRender.com.

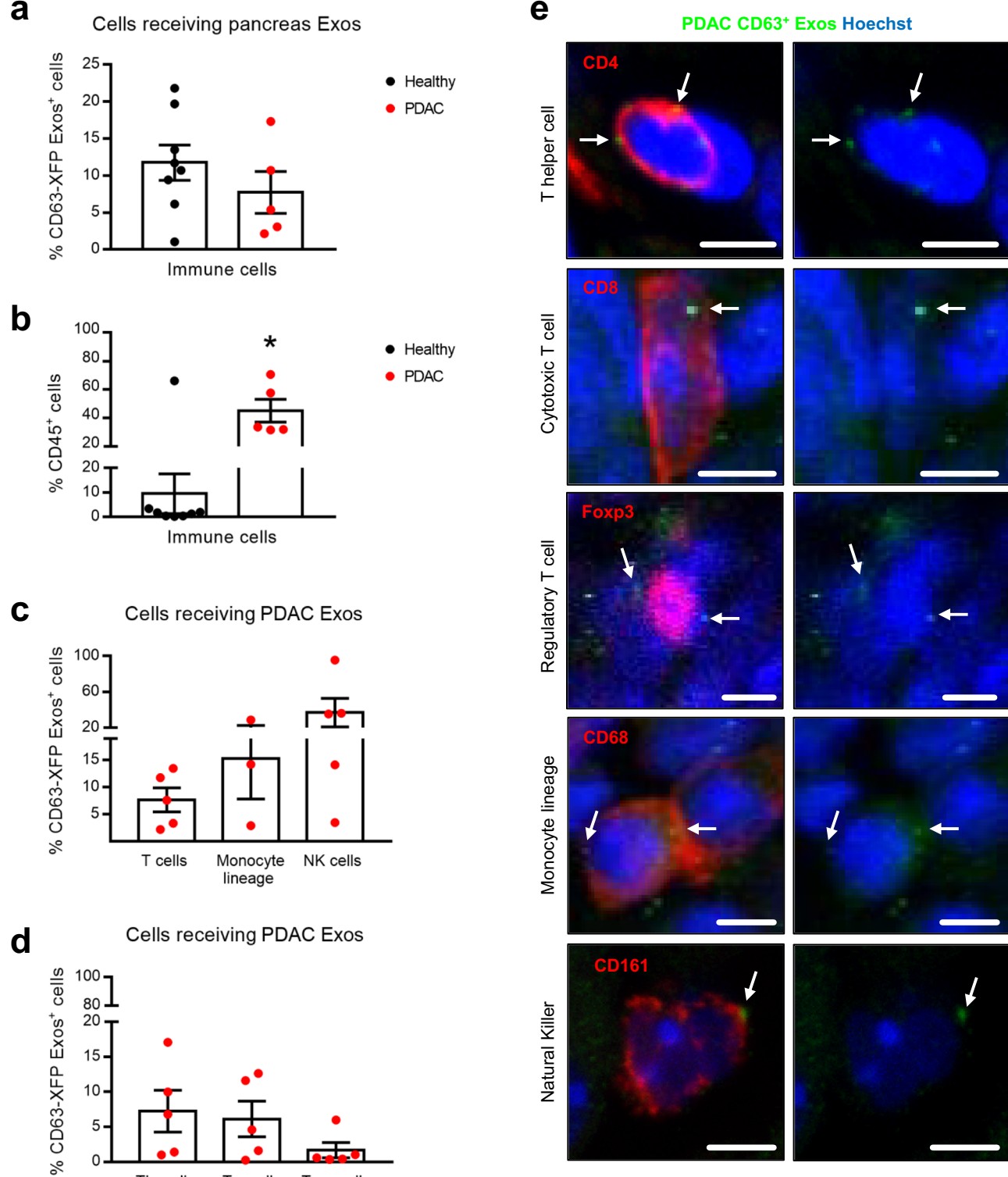

**Fig. 4 | PDAC exosomes mediate local communication with cells of the immune system.** **a** Dot plot representing the percentage of cells of the immune system (CD45⁺) that received pancreas-derived CD63⁺ Exos in health (Panc-CD63-mCherry, $n = 8$) and in PDAC (KPF CD63-mCherry $n = 2$ and KPC-ExoBow $n = 3$) analyzed by flow cytometry. **b** Dot plot representing the percentage of cells of the immune system (CD45⁺) in the pancreas microenvironment in health (Panc-CD63-mCherry, $n = 8$) and in PDAC (KPF CD63-mCherry $n = 2$ and KPC-ExoBow $n = 3$) analyzed by flow cytometry. Two-tailed Mann–Whitney test, $p = 0.0186$. **c** Dot plot representing the percentage of T cells (TCRβ⁺), cells of the monocyte lineage (CD11b⁺Ly6G/C⁻) and natural killer (CD11b⁺NK1.1⁺) cells that received PDAC CD63⁺ Exos (KPF

CD63-mCherry $n = 2$ and KPC-ExoBow $n = 3$, except for monocyte-lineage $n = 3$ KPC-ExoBow) analyzed by flow cytometry. **d** Dot plot representing the percentage of T helper cells (Th, TCRβ⁺CD4⁺), cytotoxic T cells (Tc, TCRβ⁺CD4⁻), and regulatory T cells (Treg, CD4⁺Foxp3⁺) that received PDAC CD63⁺ Exos (KPF CD63-mCherry $n = 2$ and KPC-ExoBow $n = 3$) analyzed by flow cytometry. **e** Representative confocal microscopy images of PDAC CD63⁺ Exos (green) accumulation in different subpopulations of the tumor microenvironment (red) including T helper cells, cytotoxic T cells, and regulatory T cells, cells of the monocyte lineage and natural killer cells. Nuclei were counterstained with hoechst (blue). Scale bar 5μm. Data are Mean ± SEM. Source data are provided as a Source Data file.

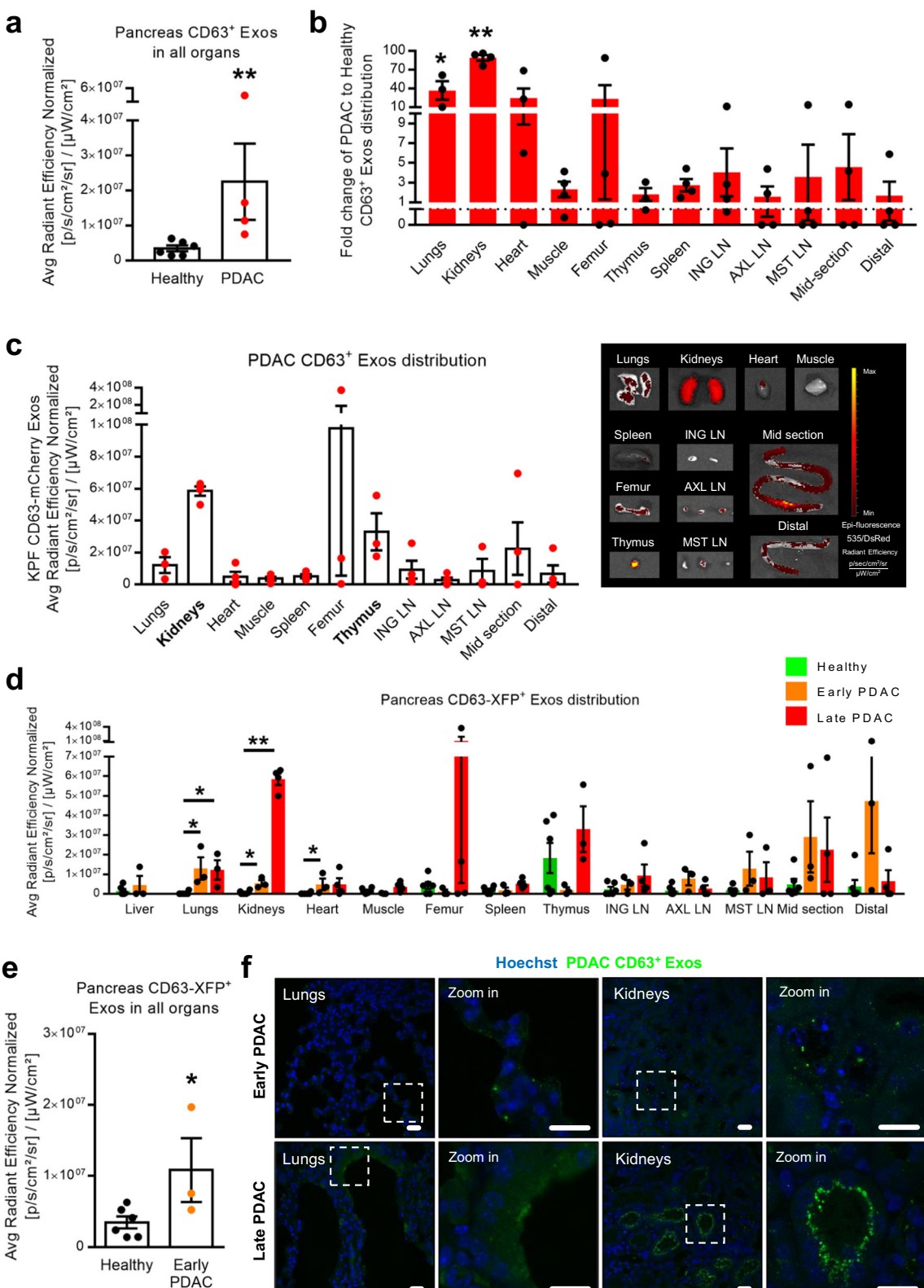

overlapping biological functions. This alignment was supported by the enrichment of angiogenesis-related proteins in both WT and KPC small EVs (Fig. 7c). However, 29% of the proteomic content in WT and KPC small EVs was distinct, with 24% of the detected proteins being specific to KPC small EVs, and only 5% specific to WT small EVs (Fig. 7b). This divergence suggested a more diverse protein repertoire in the cancer context, potentially reflecting broader biological roles, particularly in the Cell Activation pathway (Fig. 7d).

A closer examination of gene ontology analysis of the upregulated proteins detected in KPC compared to WT small EVs revealed their involvement in cell proliferation and apoptosis (Fig. 7e). To delve deeper into these small EVs populations, we conducted RNA

**Fig. 5 | Pancreas exosomes-mediated inter-organ communication increases throughout PDAC progression. a** Dot plot representing the average radiant efficiency fluorescence levels of CD63-mCherry⁺ Exos in the different organs in health (Panc-CD63-mCherry, $n = 6$) and in PDAC (KPF CD63-mCherry, $n = 4$). Two-tailed Mann Whitney, $p = 0.0095$. **b** Fold change of the average radiant efficiency fluorescence levels of CD63-mCherry⁺ Exos in the different organs in PDAC (KPF CD63-mCherry, $n = 4$) in relation to the healthy context (Panc-CD63-mCherry, $n = 6$). Two-tailed Mann-Whitney test, *$p = 0.0119$, **$p = 0.0095$. **c** Dot plot representing the average radiant efficiency fluorescence levels of PDAC CD63-mCherry⁺ Exos across different organs in KPF CD63-mCherry mice ($n = 4$) at time of euthanasia (left), with representative IVIS images (535 excitation laser and DsRed emission filter; right). **d** Dot plot representing the average radiant efficiency fluorescence levels of pancreas CD63⁺ Exos across all organs in different disease stages, healthy (Panc-CD63-

mCherry, $n = 6$), early PDAC (KPF CD63-mCherry, $n = 1$ and KPC-ExoBow, $n = 2$) and late PDAC (KPF CD63-mCherry, $n = 4$). Two-tailed Mann-Whitney test, in healthy *vs.* early PDAC: lung $p = 0.0119$, kidneys p $= 0.0238$, heart p $= 0.0238$ and healthy *vs.* late PDAC: lung $p = 0.0119$, kidneys $p = 0.0095$. **e** Dot plot representing the average radiant efficiency fluorescence levels of CD63-XFP⁺ Exos present in the different organs in health (Panc-CD63-mCherry, $n = 6$) and in early PDAC (KPF CD63-mCherry, $n = 1$ and KPC-ExoBow, $n = 2$). Two-tailed Mann–Whitney test, $p = 0.0476$. **f** Representative confocal microscopy images of PDAC CD63⁺ Exos (green) accumulation in the lungs and kidneys of KPF CD63-mCherry mice at early (upper panel) or late (lower panel) PDAC stages. Nuclei were counterstained with hoechst (blue). Scale bar 20 µm. Data are Mean ± SEM. ING LN, inguinal lymph nodes, AXL LN axillary lymph nodes, MST LN mesenteric lymph nodes. Source data are provided as a Source Data file.

sequencing (RNA Seq) analysis and found that around 17% of the genes exhibited differential expression in KPC small EVs (3% downregulated and 14% upregulated; Fig. 7f, g). We compared the list of upregulated genes with those detected in exosomes isolated from a KPC cell line, eliminating any potential contribution of non-cancer small EVs co-isolated from the tumor in the KPC small EVs sample. This gene ontology analysis implicated pathways related to metabolism and cell death regulation (Fig. 7h).

To further understand the phenotypic changes occurring upon cancer exosomes uptake in the two major cell type targeted by PDAC CD63⁺ Exos, we exposed an ex-vivo-established CAFs cell line and an endothelial cell line (bEnd.3) to cancer exosomes. For this purpose, a cancer KPF CD63-phyYFP cell line was created from a KPF CD63-mCherry tumor, as detailed in the Methods section. The expression of CD63-phiYFP was confirmed in these cells by flow cytometry analysis, and both cells and their derived exosomes were shown to carry the fusion protein through western-blot analysis (Supplementary Fig. 12a, b). Co-culture experiments confirmed the uptake of cancer exosomes in vitro (Supplementary Fig. 12c, d).

RNA Seq analysis of CAFs and endothelial cells (bEnd.3) following exposure to cancer exosomes and their non-exposed counterparts revealed a set of differentially expressed genes (Fig. 7i). Gene ontology analysis of the upregulated genes in CAFs treated with cancer exosomes highlighted pathways related to cell differentiation, increased cell adhesion, reduced cell proliferation, programmed cell death, and altered protein metabolism (Fig. 7j). These findings reflect metabolic reprogramming, known to shape distinct CAFs behaviors and impact cancer cell metabolic switch and growth capacity[31].

Moreover, pathways related to the regulation of the immune response were also identified, potentially associated with the inflammatory or antigen-presenting CAF subtypes[32,33] (Fig. 7j). In endothelial cells, gene ontology analysis of the downregulated genes following cancer exosomes treatment implicated genes related to cell adhesion, differentiation, and blood vessels formation (Fig. 7k). These findings collectively illustrate the modulatory capacity of cancer exosomes on both CAFs and endothelial cells, aligning with the observed phenotypes in the PDAC GEMMs (Figs. 2, 3).

Finally, we explored whether the observed alterations in gene expression in both CAFs and endothelial cells could be directly linked to the RNA and protein cargo of KPC small EVs. Our analysis confirmed a direct association between the RNA and protein cargo of KPC small EVs and the differentially expressed genes in CAFs and endothelial cells following exposure to cancer exosomes (Fig. 7l, m). Notably, a higher coverage was found for the upregulated differentially expressed genes in endothelial cells compared to CAFs (Fig. 7n), illustrating the direct cargo transfer from small EVs into cells, along with the downstream molecular effects of such content delivery.

## Discussion

Recent advances in the development of tailored mouse models for in vivo tracking of exosomes have opened new avenues to uncover

their significance in different biological scenarios[34–39]. Models that closely mimic the biological system can help consolidate our existing knowledge of the functions of exosomes but also uncover features of these vesicles. This improves our understanding of the overlooked homeostatic processes, but also sheds light on the pathophysiology of cancer. Consequently, these findings hold potential for improving patient care by providing insights into the biology of the disease and by advancing the state of the art in exosomes research, enabling their use as potent therapeutic vehicles or targets[40]. In this study, we developed a versatile genetic approach using a CD63-based reporter mouse model to track exosomes specifically from pancreas cells (Fig. 8). By investigating the distribution of pancreas exosomes during PDAC progression, we made several discoveries that enlighten the biological significance of exosomes in the healthy pancreas and in PDAC. One of our key findings is that the frequency of a particular cell type does not determine the communication routes that take place. This observation suggests that intercellular communication in vivo is not random but rather occurs in a coordinated manner. This aligns with our previous discoveries regarding intra-tumor communication, which showed an orchestrated flow of EVs from cancer stem cells (~6% of cancer cells), to non-stem cancer cells (>90% of cancer cells), supporting tumor growth[30]. These findings provide insights into the dynamic interplay between different cell types during cancer progression, uncovering a local and at a distance coordinated network of communication, that provides a deeper understanding of how PDAC evolves and progresses. This knowledge could pave the way for the development of targeted therapeutic strategies aimed at disrupting this communication network to impair tumor growth and improve patient outcomes.

Our work provides evidence for the role of exosomes secreted by the healthy pancreas in restraining angiogenesis. Indeed, it has been shown that during pregnancy, placenta-derived exosomes can portray pro- or anti-angiogenic cargo that affects angiogenesis in a pathophysiological state-dependent manner[41]. Hence, exosomes could be a mean to maintain overall homeostasis and organ fitness. Interestingly, our data shows that tumor angiogenesis is also modulated by PDAC exosomes, shedding light on important mechanisms driven by exosomes in PDAC to support disease progression. The role of PDAC exosomes in the TME remodeling has been widely addressed[42]. In fact, the nature and spatial distribution of CAFs were described to be related with tumor-promoting and chemoprotective functions[26]. We demonstrate that CAFs with low expression of αSMA (usually associated with an inflammatory phenotype) are enriched nearby PDAC lesions with proficient exosomes secretion, showcasing the potential role of cancer exosomes in modulating the landscape of CAFs and influencing the prognostic of PDAC tumors[28]. Finally, communication was also detected with immune cells within the healthy pancreas and PDAC, albeit at very low levels. Despite this, one cannot disregard the effect of the exosomes in those cells, nor deny that other interactions may occur in vivo through a ligand-receptor mechanism rather than endocytosis or fusion with recipient cells[9]. The integrated analysis of

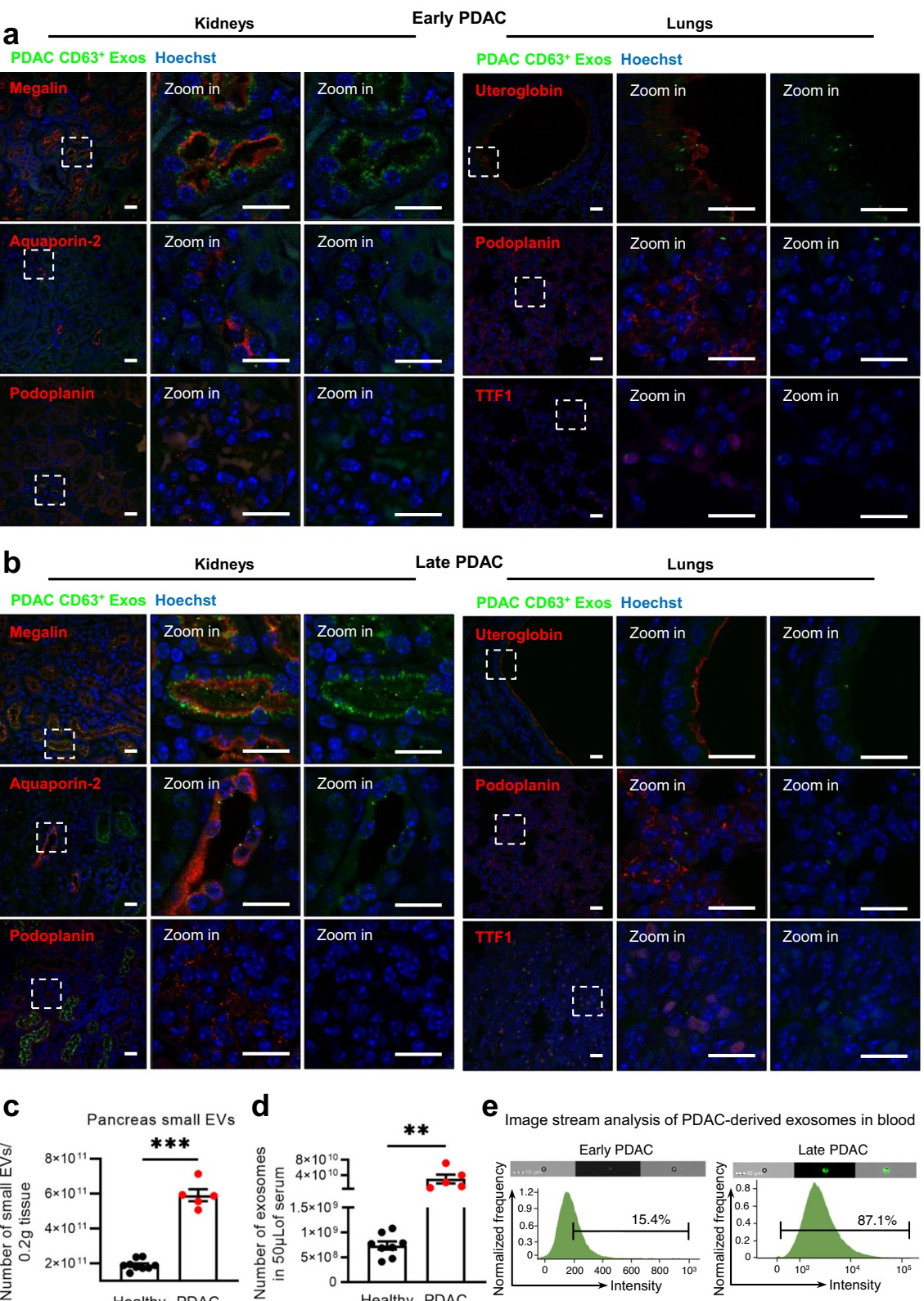

**Fig. 6 | PDAC exosomes are enriched in circulation and are taken up by specific cell types in the kidneys and lungs.** Representative confocal microscopy images of PDAC CD63+ Exos (green) accumulation in the kidneys (Megalin, Aquaporin-2 and Podoplanin positive cells in red) or lungs (Uteroglobin, Podoplanin and TTF1 positive cells in red) of KPF CD63-mCherry mice at **a** early PDAC stages or **b** late PDAC stages. Nuclei were counterstained with hoechst (blue). Scale bar 20 μm. Experiments performed in a total of 3 mice. **c** Nanoparticle tracking analysis of the small EVs population isolated from the pancreas of healthy ($n = 9$, *wild-type n = 3*, Panc-CD63-mCherry $n = 3$ and Panc-ExoBow $n = 3$) or PDAC mice (PKT, $n = 5$) according to Crescitelli et al.[53]. Two-tailed Mann–Whitney test, $p = 0.0005$. Data are Mean ± SEM. **d** Nanoparticle tracking analysis of the exosomes found in serum of healthy (*wild-type, n = 8*) or PDAC mice (PKT, $n = 5$). Two-tailed Mann–Whitney test, $p = 0.0016$. Data are Mean ± SEM. **e** Image stream analysis of CD63-XFP in exosomes isolated from serum of mice at an early PDAC stage (CD63-mTFP, left) and a late PDAC stage (CD63-mCherry, right). Experiment repeated in a total of 2 and 3 mice, respectively. Scale bar 10 μm. Source data are provided as a Source Data file.

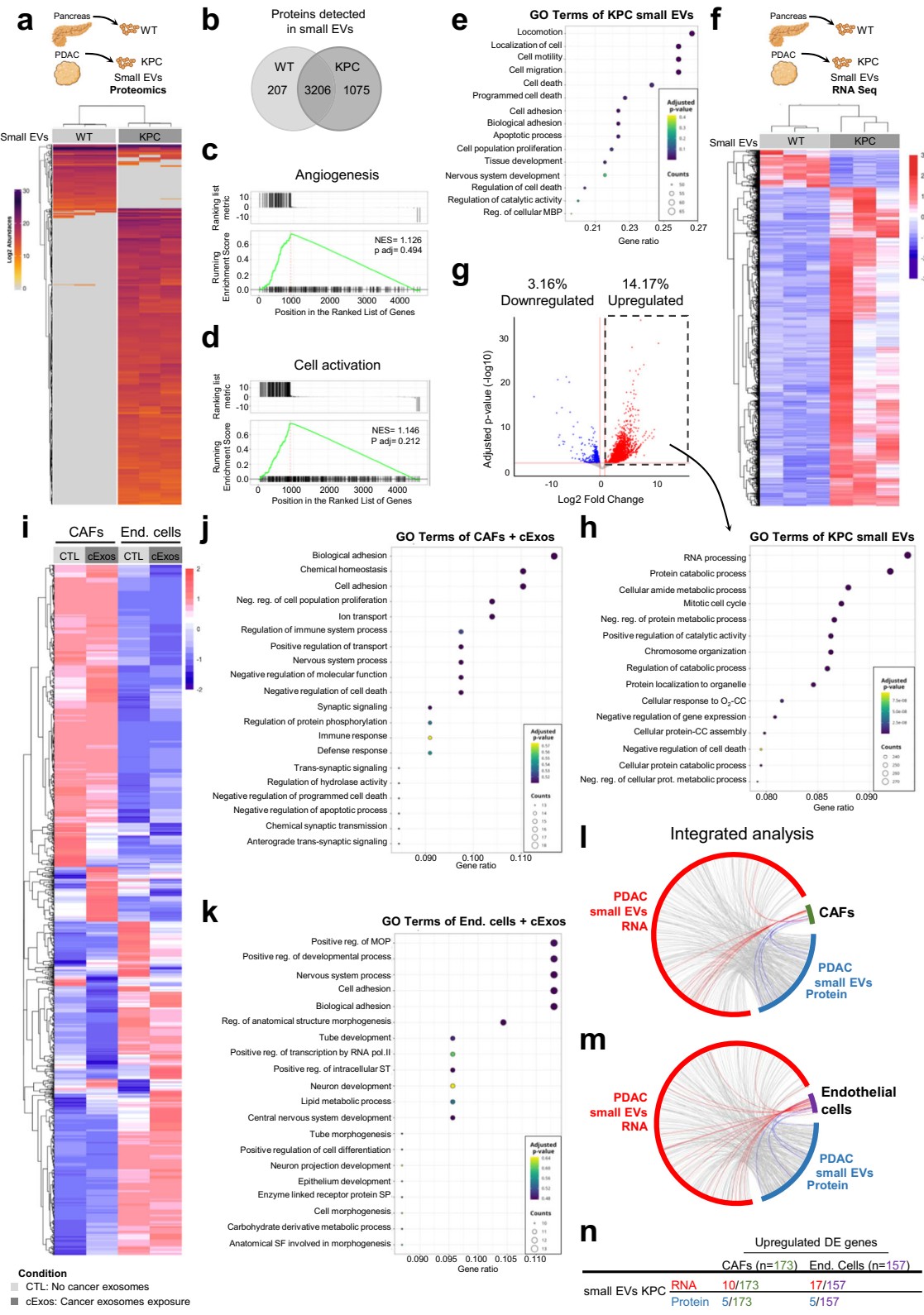

PDAC small EVs, combining both protein and RNA content, along with the changes in RNA expression in fibroblasts and endothelial cells following exposure to cancer exosomes, provide valuable insights into the mechanistic underpinnings of observed in vivo phenotypes. This multifaceted approach underscores the direct and indirect influence of EVs on target cells, ultimately resulting in their reprogramming and the remodeling of the TME. Furthermore, the concurrent increase in the abundance of small EVs within the PDAC context, compared to a

healthy environment, highlights the potential combined impact of both EVs quantity and cargo diversity in shaping the intricacies of the TME.

Inter-organ communication was enhanced in PDAC in comparison to the healthy pancreas. The kidneys, in particular proximal tubes followed by the collecting tubes, and the thymus are the organs where most cancer exosomes accumulate. The molecular mechanisms underlying the observed organotropism and its biological impact in

**Fig. 7 | EVs content reflect the local intratumor communication of PDAC.**
**a** Schematic representation of the experimental approach for proteomics analysis (upper panel) and heatmap depicting the deferentially expressed proteins common across the 3 samples for wild-type (WT) or KPC small EVs (lower panel). Unsupervised hierarchical clustering showing separation of the two protein clusters WT and KPC small EVs. **b** Venn-diagram of total proteins detected in EVs from WT or KPC small EVs. Gene set enrichment analysis of the **c** Angiogenesis pathway that does not separate the WT from KPC small EVs, and the **d** Cell activation pathway which distinguishes WT from KPC small EVs. GSEA (Gene set enrichment analysis, in **c** and **d**) uses a ranked gene list, in our case, sign(fold change gene)· −log10(P), encompassing the differential expression between two conditions (KPC vs WT), and the Kolmogorov-Smirnov statistic to score the enrichment of a priori defined set of genes that share common biological function. Significance of the score is evaluated using an empirical permutation test correcting for multiple hypothesis testing. **e** Top 15 enriched reactome pathways in KPC small EVs in comparison to WT small EVs. MBP−Macromolecule Biosynthetic process. **f** Schematic representation of the experimental approach for proteomics analysis (upper panel) and heatmap depicting the deferentially expressed genes of WT or KPC small EVs following RNA Seq analysis (lower panel). **g** Volcano plot representing the downregulated and upregulated genes in KPC small EVs in comparison to WT small EVs using DESeq2. DESeq2 uses negative binomial generalized linear models for the differential analysis of count data and uses the Wald test with

multiple correction (Benjamini−Hochberg method) for significance testing. Shrinkage of log2FC estimates to control for small sample sizes and low read counts was done by the apeglm method. **h** Top 15 enriched reactome pathways in the upregulated genes of KPC small EVs, common to KPC cell line exosomes in comparison to WT small EVs. Cellular response to oxygen-containing compound ($O_2$-CC); Cellular protein-containing complex assembly (CC). **i** Heatmap depicting the Log2FoldChange > 1 in RNA Seq analysis of cancer associated fibroblasts (CAFs) or endothelial cells (bEnd.3) exposed to cancer exosomes or not (control). **j** Top 20 enriched reactome pathways in the upregulated genes of CAFs upon cancer exosomes exposure. **k** Top 20 enriched reactome pathways in the downregulated genes of bEnd.3 upon cancer exosomes exposure. MOP Multicellular Organism Process, ST Signal Transduction, SP Signaling Pathway, SF Structure Formation. Over-Representation (ORA) analysis in (**e, h, j** and **k**) employs a hypergeometric test, corrected for multitple testing using Benjamini−Hochberg method, to determine the statistical significance of the up or down-regulated DEGs in each GO term. Circos plot depicting the interaction between the KPC small EVs RNA (red) and protein (blue) cargo with the altered genes upon cancer exosomes exposure in **l** CAFs and **m** endothelial cells (bEnd.3). In all cases are represented the upregulated differentially expressed genes or proteins. **n** Number of entries for RNA or protein identified in KPC small EVs that are present in the upregulated differentially expressed genes of CAFs or endothelial cells exposed to cancer exosomes. Source data are provided as a Source Data file. Schemes created with BioRender.com.

disease progression remains to be elucidated. Regarding communication with organs that host PDAC metastasis, cancer exosomes accumulated in the lungs (both in early and late PDAC stages), yet the same observation was not true for the liver, which is the most frequent site of metastasis[43]. Still, our observations are limited to the models used of CD63[+] Exos tracing, and other EVs subpopulations could be behind different biological roles of EVs.

Although a great effort has been made to improve the current models and methodologies to study exosomes in vivo, no model is bulletproof, and several considerations are at place. First, the nanosize of exosomes represents a major hurdle in their analysis, and is limited to the detection threshold of the techniques used. In addition, cells are described to produce heterogenous populations of exosomes and a consensus marker that includes them all, if existing, is yet to be determined. Specifically, in our work, we study the CD63[+] population of exosomes. Even though CD63 is widely described as an exosomes marker, it does not cover all endosomal-derived exosomes, and on the other hand, it can also be found in other EVs subpopulations[13,44]. Although the use of a broad exosomes marker might sound compelling, the heterogenous nature of these vesicles might reflect their biological role in vivo, which would be lost in such approach. Indeed, the most suitable tetraspanin of choice may vary between tissue/cell-type or even according to the pathophysiological context. Hence, only a comprehensive and coordinated analysis using different models in the same biological setting will provide valuable information to the questions that remain elusive. Furthermore, the development of in vivo models with different reporter systems, in addition to the fluorescent ones as here described will be of great value to enable the use of different techniques to address the biodistribution of exosomes[45]. Nonetheless, despite the latest sophisticated strategies to label and trace injected EVs in vivo, it is clear that the distribution of these vesicles is dependent on several factors[46]. With the cell source used, the amounts injected, the route of administration, the timepoint of analysis after injection, and the immunosuppressive setting of the experimental mouse among the most critical ones[47]. Hence, despite the usefulness of such techniques, new approaches that enable the study of EVs in a context that closely resembles physiological conditions provides valuable information to uncover the biological significance of communication mediated by EVs in vivo. In most studies where EVs are administered into mice, they end up in the liver, lungs, spleen, and kidneys[48]. As reported, these were not identified as the major target sites of communication in the healthy pancreas, but in

PDAC, the lungs and kidneys were indeed identified. Nevertheless, one might not disregard the potential effect of overexpressing an exosomes marker instead of investigating the biodistribution when using the endogenous expression levels of the tagged vesicles.

Altogether, we have mapped the intra-pancreas and inter-organ distribution of exosomes, in the healthy pancreas and PDAC. We demonstrate that communication is not a random event but rather depends on the biological context rather than the prevalence of distinct populations of cells, and that these specific routes of communication that take place in vivo influence the composition of the pancreas microenvironment with a specific impact on angiogenesis and spatial distribution of CAFs (Fig. 8). Additionally, we have developed a GEMM to tag and trace exosomes in a lineage-specific manner that can be used in a multitude of biological settings. The broad use of this model in other contexts, pathological and non-pathological, will elucidate on whether the communication routes governed by CD63[+] exosomes are specific to the pancreatic context or if more transversal mechanisms are at play. The potential use of this model in its multi-color form further strengthens its versatility and relevance. Dissecting communication routes between subpopulations of the same cell of origin could be crucial to further understand processes such as homeostasis and intratumor heterogeneity.

We believe this study contributes to a better understanding of the biological significance of pancreas derived exosomes in health and disease.

## Methods
### Cell culture
We have used the following human PDAC cell lines: T3M4 (RCB Cat# RCB1021 kindly provided by Dr. Christoph Kahlert, Universitatsklinikum Carl Gustav Carusan der Technischen Universitat Dresden, Germany), PANC-1 (ATCC Cat# CRL-1469), BxPC-3 (ATCC Cat# CRL-1687), and MIA PaCa-2 (ATCC Cat# CRL-1420). Cell lines were cultured in RPMI-1640 medium (Gibco) supplemented with 10% (v/v) fetal bovine serum (FBS, Gibco), 100 U/mL penicillin and 100 µg/mL streptomycin (Gibco) and STR profiled. All stable clones derived from the parental BxPC-3 and MIA PaCa-2 cell lines were cultured in the same conditions. In addition, a mouse endothelial cell line bEnd.3 (ATCC Cat# 2299) cultured in DMEM media (Thermo Scientific 10566016) supplemented with 1 mM sodium pyruvate (VWR HYCLSH30239.01), 10% (v/v) fetal bovine serum (FBS, Gibco), 100 U/ mL penicillin and 100 µg/mL streptomycin (Gibco) was used. All cell

## Tracing of endogenously produced pancreas-CD63⁺ exosomes *in vivo*

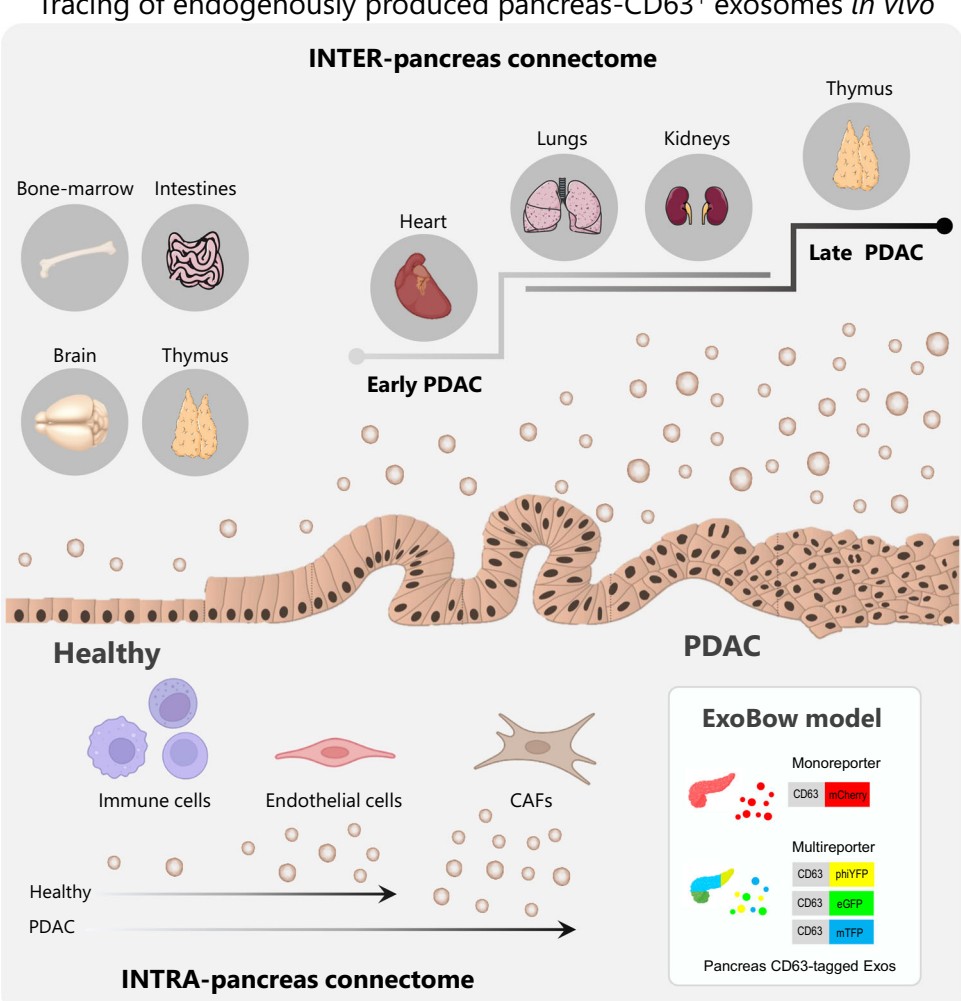

**Fig. 8 | Schematic of the intra- and inter- pancreas communication established by CD63 positive exosomes in PDAC.** The ExoBow mouse model can be crossed with specific Flp and Cre lines driven by the *Pdx1*-pancreas promoter which will render the conditional expression of CD63-XFP reporter proteins by pancreas cells. Crossing this model with well-stablished PDAC genetically engineered mouse models, enables the assessment of CD63 exosomes biodistribution locally and systemically in both healthy and PDAC contexts. Our work demonstrates an intra-pancreas connectome mediated by pancreas-derived exosomes mainly with CAFs (in a PDAC-specific context), followed by endothelial cells and, in lower amounts, with cells of the immune system. We were also able to demonstrate that the inter-pancreas connectome mediated by pancreas-derived exosomes varies from physiological conditions to a cancer context. Furthermore, we show that these routes of communication also vary along PDAC progression. PDAC, pancreatic ductal adenocarcinoma; CAFs, cancer-associated fibroblasts. Schemes created with BioRender.com.

---

lines were routinely mycoplasma tested and maintained at 37 °C in a humidified chamber with 5% $CO_2$.

### Cell line transfection
BxPC-3 cells were separately transfected with the following CD63-XFP vectors: CD63-mCherry, CD63-phiYFP, CD63-eGFP and CD63-mTFP cloned into pRP[Exp]-Puro-CAG plasmid backbone in collaboration with Vector Builder Inc. Reverse transfection of each plasmid (2.5 μg DNA/1.5 ×10⁵ cells) was performed using Invitrogen Lipofectamine® 2000 (Thermo Fisher Scientific) according to the manufacturer's instructions. To obtain stable clones, puromycin (1 μg/mL, Sigma-Aldrich P8833) selection and fluorescence activated cell sorting (FACS) based on the expression of fluorescent proteins was performed. MIA PaCa-2 CD63-GFP cell line was established as previously described by Ruivo et al.[30].

### Primary cell culture establishment
Primary cultures of cancer cells or CAFs derived from PDAC tumors were cultured in RPMI-1640 medium (Gibco) supplemented with 20%

(v/v) fetal bovine serum (FBS, Gibco), 100 U/mL penicillin and 100 μg/mL streptomycin (Gibco). Briefly, tumors were minced and digested in digestion buffer (0,012% Dispase II Sigma-Aldrich D4693, 0,012% Collagenase Sigma-Aldrich C7657 in Hanks Balanced Salt Solution−HBSS 1 X) for 20 min at 37 °C in agitation. Next, cells were filtered (70 μm strainer, Falcon) and incubated with Red Blood Cell lysis buffer for 5 min at 37 °C, followed by a washing step. Single-cell suspension was then added to a cell culture plate. Differential trypsinization was performed to enrich each sample in cancer cells or CAFs[49]. CAFs derived from a KPC tumor. Primary cancer cells expressing CD63-mCherry were obtained from a KPF-CD63-mCherry tumor. Primary cancer cells expressing CD63-phiYFP resulted from the transfection of cancer cells established from a KPF-CD63-mCherry mouse with the pCAG-Cre (Addgene #13775). CD63-mTFP and CD63-eGFP expressing cancer cells were obtained from the transfection of cancer cells established from a KPC-ExoBow Flp negative tumor with pSICO-Flpo (Addgene #24969). In both cases using 2.5 μg DNA/1.5 × 10⁵ cells and, after transfection, single cell sorting based on FITC expression was performed to a 96 multiwell plate followed by clone expansion. PCR screening of the

clones was used to identify the fusion protein (CD63-XFP) recombined for in each one, using the primers described in section Polymerase chain reaction (PCR). All cell lines were routinely mycoplasma tested and maintained at 37 °C in a humidified chamber with 5% $CO_2$.

## Exosomes isolation from cell culture medium and sucrose gradient

BxPC-3 cells were cultured in RPMI-1640 medium (Gibco) supplemented with 10% FBS (Gibco) depleted of EVs by overnight ultracentrifugation at 100 000 g, and 100 U/mL penicillin and 100 μg/mL streptomycin (Gibco). After 72 h, the medium was collected and centrifuged at 2500 g for 10 min followed by a 5 min centrifugation at 4000 g. Subsequently, the medium was filtered through a 0.2 μm filter (GE Healthcare Whatman™) directly to an ultra-clear centrifuge tube (Beckman Coulter®). The samples were centrifuged overnight at 100,000 g, 4 °C using the Optima™ L-80 XP ultracentrifuge, Beckman Coulter. The supernatant was carefully removed, and the isolates were subjected to a continuous sucrose gradient (0.25–2 M) as described in[50]. Briefly, the pellet was resuspended in 2 mL of HEPES/Sucrose stock solution (HEPES 20 mM/protease-free sucrose 2.5 M, pH 7.4) and transferred to an ultra-clear centrifuge tube. A continuous sucrose solution (2 M to 0.25 M) was dispensed into the ultracentrifuge tube containing the EVs suspension. Samples were centrifuged overnight (>14 h) at 210,000 g, 4 °C. After ultracentrifugation, 1 mL of gradient fractions, from top to bottom, were collected. 50 μL of each fraction was used to measure the refractive index in a refractometer. Each fraction was individually placed in an ultra-clear centrifuge tube, diluted in NaCl 0.9% and centrifuged at 100,000 g for 2 h, at 4 °C. The subsequent pellet was resuspended in 30 μL 2.5% SDS/8 M Urea for protein extraction and incubated for 30 min on ice, followed by a 30 min centrifugation at 17,000 g, 4 °C. The supernatant was stored at −20 °C.

## Size-exclusion chromatography

BxPC-3 EVs isolated via ultracentrifugation (as previously described) were further purified by size-exclusion chromatography using the automatic fraction collector (Izon). Briefly, a qEV original column 35 nm Gen2 (Izon) was used to separate 500 μL of EVs preparation in 13 fractions of 400 μL with a buffer volume of 2.1 mL. For downstream analysis, each fraction but fraction 1 which was discarded, was ultracentrifuged at 100,000 g for 3 h. Fractions were analyzed by western-blot as described in detail in the western-blot section.

## Image stream

Exosomes were collected from 10 mL of T3M4, PANC-1, and BxPC-3 cell supernatant cultured for 72 h in RPMI medium supplemented with EV-depleted FBS or from mice serum samples. EVs isolation was performed by ultracentrifugation as previously described. The exosomes pellet was resuspended in phosphate-buffered saline (PBS) 1X and incubated with aldehyde/sulfate 4 μm beads (Thermo Fisher Scientific A37304) for 45 min in rotation. Next, 1 M glycine was added and incubated in rotation at room temperature (RT) for 1 h. After centrifugation at 13,800 g for 2 min, supernatant was discarded, and beads were resuspended in bovine serum albumin (BSA) 10% and incubated for 45 min in rotation. Beads were then resuspended in 20 μL of BSA 2%. Half of the sample was used for incubation with primary antibody and secondary antibody, while the other half was used for incubation with secondary antibody only (control). Incubation with anti-CD63 antibody was performed overnight at 4 °C and in rotation (1:400, Santa Cruz Biotechnology sc-15363). On the following day, washing steps were performed using BSA 2%. Next, incubation with secondary antibody was performed for 30 min at RT with goat anti-rabbit IgG Alexa Fluor 488 (1:500, Thermo Fisher Scientific A-11034).

For exosomes serum samples, staining of intraluminal exosomal proteins followed the adapted protocol from Kugeratsk et al.[51]. Briefly,

exosomes pellet was resuspended in 100 μl of PBS 1X and mixed with 6 μl of aldehyde/sulfate beads (Thermo Fisher Scientific A37304) that had been previously diluted in 50 μl of PBS 1X. After 15 min incubation at RT, samples were left overnight in rotation at 4 °C. The following day, 150 μl of 1 M glycine was added and incubated at RT for 1 h in rotation. Samples were centrifuged for 2 min at 13 800 g and supernatant discarded. Blocking was achieved using 100 μl of 10% BSA and incubating for 1 h at RT while rotating. Then, samples were centrifuged and a fixation/permeabilization step performed for 30 min on ice using 100 μl of Fixation/Permeabilization reagent (eBioscience 00-5123-43 and 00-5223-56). After which samples were split into two (one being the negative control with only secondary antibody incubation) and washed with 200 μl of 1X Permeabilization buffer (eBioscience 00-8333-56). Next, antibodies incubation was performed in 25 μl of 2% BSA/ 1X Permeabilization buffer for 1 h on ice. The antibodies were used at 1:100 dilution, rat anti-TFP and rabbit anti-mCherry (kindly provided by Cai Laboratory, University of Michigan Medical School, Michigan, USA), Samples were washed 3 times followed by secondary antibody incubations with anti-rabbit Alexa-Fluor® 488 (1:200, Jackson ImmunoResearch, 711-545-152) or anti-rat Alexa-Fluor® 488 (1:200, Invitrogen, A21208). Next, samples were washed 3 times and resuspended in 500 μl of 1X Permeabilization buffer for flow cytometry analysis.

Detection was made using the ImageStream^x MarkII Imaging Flow Cytometry (Luminex Amnis Image Stream Multispectral Imaging Flow Cytometer). The nanoparticle tracking analysis (NTA) (NanoSight NS300) was used to determine EVs concentration.

## Production of CAG/mouseCD63-XFP GEMM using embryonic stem cells

The R26^CD63-XFP/+ allele was developed in collaboration with Cyagen. Briefly, the "CD63-cDNA-LoxN-Lox2272-Lox5171-mCherry-bGHPolyA-LoxN-phiYFP-rBGPolyA-Lox2272-GFP-TKPolyA-Lox5171-mTFP-PGKPolyA" transgene was cloned into intron 1 of ROSA26 and the "CAG-Frt-Stop (Neo cassette with 3*SV40 PolyA)-Frt" was placed upstream of the transgene. The model was generated by homologous recombination in embryonic stem cells.

The diphtheria toxin (DTA) cassette, the ROSA26-homology arms, the CAG promoter (CMV enhancer, chicken beta-Actin promoter and rabbit beta-Globin splice acceptor site), and the bGH polyadenylation have been cloned into backbone as prepared. To engineer the targeting vector, the components of neo-3*SV40pA, mcherry-bGHpA, phiYFP-rBGpA, EGFP-TKpA, and mTFP-PGKpA were generated by PCR using gene synthesis product as template. PCR primers were designed to share 15–20 bases of homology with the sequence at the end of the linearized backbone vector.

Mouse genomic fragments containing homology arms (HAs) and conditional knock-in region were amplified from BAC clone (RP23-244D9) by using high fidelity Taq. Next, the fragments were sequentially assembled into a targeting vector together with recombination sites and selection markers. Each individual cloning step was extensively validated through restriction analysis and partial sequencing.

In detail, the neo-3*SV40pA cassette was cloned into Basic Vector by In-Fusion Enzymes, and the Basic Vector came from digested products of NsiI/XhoI. The correct plasmid was named Cd63-Step1. The Cd63-cDNA-3*loxp (NCBI Reference Sequence: NM_007653.3; Ensembl: ENSMUSG00000025351; Transcript: Cd63-201 ENSMUST00000026407.7) were cloned into Cd63-Step1 by In-Fusion Enzymes, and the Cd63-Step1 came from digested products of KpnI/XhoI. The correct plasmid was named Cd63-Step2. The mcherry-bGHpA & phiYFP-rBGpA cassette were cloned into Cd63-Step2 by In-Fusion Enzymes, and the Cd63-Step2 came from digested products of ClaI/AsiSI. The correct plasmid was named Cd63-Step3. The EGFP-TKpA & mTFP-PGKpA cassette were cloned into Cd63-Step3 by In-Fusion Enzymes, and the Cd63-Step3 came from digested

products of AsiSI/XhoI. All steps were subsequently confirmed to be the correct targeting vector by diagnostic PCR, restriction digests and sequencing. This plasmid was subsequently linearized with NotI and used for electroporation to ES cells (C57BL/6).

Negative selection was performed by the expression of diphtheria toxin A (DTA) (present upstream of the HA of the targeting vector), reducing the isolation of non-homologous recombined ES cell clones. Further positive selection with neomycin drug led to 95 drug-resistant clones. PCR screening confirmed 15 potentially targeted clones, 6 of which were validated by Southern Blotting. Next, selected ES cells were used for blastocyst microinjection, followed by chimera production.

For reference, each fluorescent reporter ends with a polyA sequence that enhances expression by stabilizing the transcript and preventing its degradation[52].

### Embryonic stem cells (ESC) transfection
ESC containing the ExoBow transgene ($2 \times 10^5$ cells) were reversely transfected with 2.5 µg of each plasmid using LP300. EGFP plasmid was used as a positive control of transfection. After transfection, the mixture was plated in 6 well plates. At the end of the day (after 5 h of transfection), the medium was replaced with fresh ES cell medium. After 72 h, PCR analysis was performed to evaluate transgene recombination.

### Mice
Equal number of female and male $Rag2^{-/-}Il2rg^{-/-}$ (Cat# JAX:014593) mice were orthotopically injected in the pancreas with $2.0 \times 10^6$ cells derived from MIA PaCa-2 or MIA PaCa-2 CD63-GFP cell lines as previously described[30]. Tumor growth was monitored by ultrasound (Micro Ultrasound Vevo 2100). Mice were euthanized when the tumor reached 1500 mm³ or when presented with severe symptoms. Panc-CD63-mCherry and Panc-ExoBow mice between 8–11 weeks of age were used in all experiments. Pdx1-Flp alleles were kindly provided by Dr. Dieter Saur, Technische Universität München, München, Germany and Dr. Michael Ostrowski, Department of Biochemistry and Molecular Biology, Medical University of South Carolina, Charleston, USA[18,19]. KPF CD63-mCherry ($FSF\text{-}KRAS^{G12D/+}$; $TrpS3^{Frt/+}$; $Pdx1\text{-}Flp$; $R26^{CD63\text{-}XFP/+}$) developed spontaneous PDAC tumors in a similar way to the KPF mice[18]. All alleles of the KPF mouse model were kindly provided by Dr. Dieter Saur, Technische Universität München, München, Germany. Mice were sacrificed from 15–21 weeks of age. KPC-ExoBow Flp negative mice ($LSL\text{-}Kras^{G12D/+}$; $LSL\text{-}Trp53^{R172H/+}$; $Pdx1\text{-}Cre$; $R26^{CD63\text{-}XFP/+}$) developed spontaneous PDAC tumors in a similar way to the KPC mice. KPC ($LSL\text{-}Kras^{G12D/+}$; $LSL\text{-}Trp53^{R172H/+}$; $Pdx1\text{-}Cre$) alleles were purchased from Jackson Laboratory: B6.129 S4-Krastm4Tyj/J (IMSR Cat# JAX:008179); 129S-Trp53tm2Tyj/J (IMSR Cat# JAX:008652) and B6.FVB-Tg(Pdx1-cre)6Tuv/J (IMSR Cat# JAX:014647). In the KPC-iExoBow model, the R26$^{LSL\text{-}FLPoERT2/+}$ was kindly provided by Dr. David Goodrich, Roswell Park Cancer Institute, Buffalo, USA. Tamoxifen treatments were performed upon detection of a palpable tumor mass through intraperitoneal injection of 100 µL of 20 mg/mL tamoxifen (Sigma-Aldrich T5648) diluted in corn oil tamoxifen. Mice were sacrificed from 17–24 weeks of age. KPC-ExoBow Flp negative mice were orthotopically injected with $1 \times 10^{11}$ GCU adeno-associated virus 8 (AAV8) Vector Biolabs (Cat#:1775) at 6 weeks of age. Surgery protocol was performed as previously described[30]. Mice were sacrificed from 8–12 weeks of age. The Rab27a$^{Frt/Frt}$ was bred with Pdx1-Flp to obtain Pdx1-Flp Rab27aKO mice that were sacrificed at around 8 weeks of age for pancreas collection, processing and analysis[18,30]. The PKT alleles ($Ptf1a\text{-}Cre$; $LSL\text{-}Kras^{G12D/+}$; $Tgfbr2^{loxP/loxP}$) were purchased from Jackson Laboratory: Ptf1atm1(cre)Hnak/RschJ (RRID:IMSR_JAX:023329); B6;129-Tgfbr2tm1Karl/J (RRID:IMSR_JAX:012603). The PKT iRab27a ($Ptf1a\text{-}Cre$; $LSL\text{-}Kras^{G12D/+}$; $Tgfbr2^{loxP/loxP}$; $R26^{LSL\text{-}FLPoERT2/+}$; $Rab27a^{Frt/Frt}$) developed spontaneous PDAC tumors in a similar way to the PKT mice. Tamoxifen treatments to induce recombination were given to pups via lactation through oral gavage of the mother with 6 mg of tamoxifen diluted in corn oil (20 mg/mL, Sigma-Aldrich T5648) at days 0, 1, 2 and 4 post-birth. PKT Rab27a-tamoxifen treated mice ($Ptf1a\text{-}Cre$; $LSL\text{-}Kras^{G12D/+}$; $Tgfbr2^{loxP/loxP}$; $Rab27a^{Frt/Frt}$) was used as control. Both PKT models were sacrificed and analyzed at humane endpoint (6–13 weeks of age). Given the high genetic complexity of the GEMMs, mice were used according to availability with no particular consideration for the sex.

All mice were housed under standard housing conditions at the i3S animal facility, and all animal procedures were reviewed and approved by the i3S Animal Welfare and Ethics Body and the animal protocol was approved by DGAV "Direção Geral de Alimentação e Veterinária" (ID 015225).

### Polymerase chain reaction (PCR)
For mice genotyping, an ear fragment was digested at 56 °C for 2 h on a thermal-shaker with lysis buffer [10 mM Tris (pH 7.5), 400 mM NaCl, 2 mM EDTA (pH 8.0)] (pH 7.3–7.5), 20% sodium dodecyl sulfate (SDS) (Merck 428018), and 20 µL of 20 mg/ml proteinase K (Ambion RNA by Life Technologies AM2548). Then, 6 M NaCl was added to the extraction mixture, samples were mixed thoroughly by vortexing for 10 s, followed by centrifugation at 17,000 g for 15 min to precipitate the residual cellular debris. The supernatant was transferred to a clean tube and 100% ethanol was added to each sample, mixed thoroughly by vortexing for 10 s, and centrifuged at 17,000 g for 5 min to pellet the DNA. The DNA pellets were washed with 70% ethanol, followed by centrifugation at 17,000 g for 5 min. The pellets were completely air dried and resuspended in sterile nuclease-free water. For conventional PCR we used a commercial master mix 2x My Taq HS Mix (Bioline Bio-25046). PCR amplifications were carried out in the T-100 Thermal Cycler (Biorad). All assays included a no-template control (contained all reaction components except the genomic DNA). Oligonucleotide sequences used for genotyping were:

R26$^{CD63\text{-}XFP/+}$ Forward—CAAAGCTGAAAGCTAAGTCTGCAG
R26$^{CD63\text{-}XFP/+}$ Reverse 1—GGGCCATTTACCGTAAGTTATGTAACG
R26$^{CD63\text{-}XFP/+}$ Reverse 2—GCCATTTAAGCCATGGGAAGTTAG

The amplification protocol included an initial denaturation and enzyme activation at 95 °C for 5 min followed by 30 cycles of denaturation at 95 °C for 30 s, annealing at 55 °C for 1 min and 30 s, and extension at 72 °C for 30 s and a final extension at 60 °C for 30 s.

For ExoBow transgene recombination assessment in pancreas frozen samples, DNA extraction protocol was the same as described above.

The following primers were used for the different regions associated with Flp or Cre-mediated recombination (Supplementary Figs. 1 and 3).

Region 1—FLP recombination Primers for Region 1 (Annealing Temperature 60.0 °C):
Forward: TGCCTTTTATGGTAATCGTGCGAG
Reverse: CCCACAAAGGCCACCAGGAAGAG
Region 2—LoxN recombination Primers for Region 2 (Annealing Temperature 60.0 °C):
Forward: CTTGCTGCATCAACATAACTGTGG
Reverse: TCCATCTCCACCACGTAGGGGATC
Region 3—Lox 2272 recombination Primers for Region 3 (Annealing Temperature 60.0 °C):
Forward: CTTGCTGCATCAACATAACTGTGG
Reverse: CGTTTACGTCGCCGTCCAGCTCG
Region 4—Lox5171 recombination Primers for Region 4 (Annealing Temperature 60.0 °C):
Forward: CTTGCTGCATCAACATAACTGTGG
Reverse: ATTCACGTTGCCCTCCATCTTCAG

### In vivo imaging system—IVIS
Organs were collected and imaged on the IVIS Lumina iii (Caliper) to determine the fluorescence intensities with the appropriate excitation

and emission filters. To detect mCherry signal, 535 laser and DsRed emission filter was used, whilst for eGFP, phiYFP and mTFP, 465 laser and GFP emission filter was used. Fluorescence intensity is represented by a multireporter scale ranging from red (least intense) to yellow (most intense). In each acquisition a control organ derived from a non-reporter mouse was used to normalize the fluorescent levels of the reporter mice. For that, regions of interest (ROIs) for each organ were made using the IVIS Living image Software. For each organ, to the average radiant efficiency [p/s/cm²/sr] / [µW/cm²] value of the reporter mouse was subtracted the value of the non-reporter mouse. Signal intensity images were superimposed over gray scale reference photographs using IVIS Living image Software.

## Flow cytometry

For the establishment of the KPF-CD63-mCherry cell line, in addition to differential trypsinization, enrichment in the CD63-mCherry positive cell population was achieved based on PE-Texas-Red expression. For the establishment of the KPC-ExoBow Flp negative cell line the sorting of EpCAM positive cells was performed for the. Briefly, single-cell suspensions of PDAC cell lines were blocked for 15 min on ice with blocking buffer (FBS 10% in PBS 1X). Next, cells were centrifuged at 300 g for 5 min and incubated for 30 min on ice with anti-EpCAM-PE (1:300, BD Biosciences Cat# 563477) in staining buffer (FBS 2% in PBS 1X). Cells were washed 3 times with PBS 1X and filtered through a 35 µm cell strainer prior to cell sorting on BDFACS Aria II Cell Sorter (BD Biosciences).

Pancreas derived from Panc-CD63-mCherry mice and tumors derived from KPC-ExoBow or KPF-CD63-mCherry mice were chopped and digested in digestion buffer (0,012% Dispase II Sigma-Aldrich D4693, 0,012% Collagenase Sigma-Aldrich C7657 in HBSS 1X) for 10 min (pancreas) and 20 min (tumors) at 37 °C in agitation. After, blocking buffer (FBS 10% in HBSS 1X) was added to stop digestion and cells were filtered through a 70 µm strainer (Falcon) to obtain a single-cell suspension and centrifuged at 600 g for 5 min. Then, cells were incubated with Red Blood Cell lysis buffer for 5 min at RT and HBSS 1X was added in excess to stop the reaction. After centrifugation, a blocking step using blocking buffer was performed for 15 min at RT. Finally, antibody staining was performed in FACS buffer (FBS 2% in HBSS 1X) for 30 min at 4 °C, followed by several washing steps. Prior to analysis on BDFACS Aria II Cell Sorter (BD Biosciences) or BD FACS-CANTO II (BD Biosciences) samples were filtered through a 35µm cell strainer (Falcon).

Viable dye was used to exclude dead cells by negative gating (Fixable Viability Dye eFluor 780, 1:100 000 eBioscience Cat# 65-0865-14).

Healthy−Panc-CD63-mCherry: CD31-PECy7 (1:500, BioLegend Cat# 102523); Mix 1: CD45.2-FITC (1:200, BioLegend Cat# 104), CD11b-PerCP (1:400, Biolegend Cat# 101229), and Ly6G/C-APC (1:800, Thermo Fisher Scientific Cat# 17-5931-82); Mix 2: NK1.1-APC (1:1000, BioLegend Cat# 108710), CD3-FITC (1:300, Thermo Fisher Scientific Cat# 11-0032-82).

PDAC−KPF CD63-mCherry: CD31-PECy7 (1:500, BioLegend Cat #102523); CD140A-APC (1:200, BioLegend Cat# APA5); CD45.2-FITC (1:200, BioLegend, Cat# 104); NK1.1-APC (1:1000, BioLegend Cat# 108710). Mix 1: TCRb-APC780 (1:400, Thermo Fisher Scientific Cat# 47-5961-82), CD4-PerCP710 (1:1000, Thermo Fisher Scientific Cat# 46-0042-82) or CD4-FITC (1:200, BioLegend Cat# 100405). Mix 2: CD4-PerCP710 (1:1000, Thermo Fisher Scientific Cat# 46-0042-82). Samples were then fixed and permeabilized according to the manufactures protocol (Thermo Fisher Scientific, Foxp3/Transcription Factor Staining Buffer Set, Cat# 00-5523-00) and stained with Foxp3−APC (1:200 Thermo Fisher Scientific Cat# 17-5773-82).

PDAC−KPC-ExoBow: CD31-PECy7 (1:500, BioLegend Cat #102523); CD140A-APC (1:200, BioLegend Cat# APA5). Mix 1: NK1.1-PE (1:100, Thermo Fisher Scientific Cat# 12-5941-82), CD11b-PerCP (1:400,

Biolegend Cat# 101229), Ly6G/C-APC (1:800, Thermo Fisher Scientific Cat# 17-5931-82). Mix 2: TCRβ-PE (1:200, Thermo Fisher Scientific Cat# 12-5961-83), CD4-PerCP eFluor710 (1:1000, Thermo Fisher Scientific Cat# 46-0041-82). Mix 3: CD4-PerCP eFluor710 (1:1000, Thermo Fisher Scientific Cat# 46-0041-82). Samples were fixed and permeabilized according to the manufactures protocol (Thermo Fisher Scientific, Foxp3/Transcription Factor Staining Buffer Set, Cat# 00-5523-00) and stained with Foxp3-APC (1:200, Thermo Fisher Scientific Cat# 17-5773-82). Mix 4: Propidium Iodide (Clontech S0957), CD45-APC-Cy7 (1:1000, BD Biosciences Cat# 560579).

For cell lines evaluation of endogenous CD63-XFP expression, single-cell suspensions of PDAC cell lines were washed once with PBS 1X and filtered through a 35 µm cell strainer prior to flow cytometry analysis.

For staining of intraluminal exosomal proteins, exosomes from 10 mL of BxPC-3 cell lines were isolated by ultracentrifugation as abovementioned and the following protocol was adapted from Kugeratsk et al.[51] and described above in the Image Stream section. The antibodies were used at 1:100 dilution, anti-GFP (BIO-RAD 4745-1051), anti-phiYFP (Evrogen AB603), rat anti-TFP and rabbit anti-mCherry (kindly provided by Cai Laboratory, University of Michigan Medical School, Michigan, USA). The secondary antibodies used were sheep Alexa-Fluor® 488 (1:300, Jackson ImmunoResearch, 713-545-003), anti-rabbit Alexa-Fluor® 488 (1:200, Jackson ImmunoResearch, 711-545-152) or anti-rat Alexa-Fluor® 488 (1:200, Invitrogen, A21208). Analysis was performed on BD LSRFortessa (BD Biosciences).

Data from flow cytometry acquisition was analyzed using the FlowJo software (version 10, BD). Staining on UltraComp eBeads Compensation Beads (Thermo Fisher Scientific, Cat# 01-2222-41) according to the manufacturer's instructions was used for matrix compensation on FlowJo software.

## Isolation of EVs from pancreas tissue

Pancreas-derived EVs were isolated according to Crescitelli et al.[53]. Briefly, pancreas were collected and minced into 1−2 mm fragments, followed by 30 min incubation in agitation in RPMI-1640 (Gibco) media supplemented with collagenase D (2 mg/ml, Sigma-Aldrich) and DNase I (40 U/ml, Thermo Fisher Scientific). After enzymatic digestion, samples were meshed through a 70µm strainer (Falcon) using HBSS 1X. Samples were centrifuged at 300 g for 10 min and 2000g for 20 min to remove cells and tissue remains. Cleared supernatants were centrifuged at 16 500 g for 20 min and 118 000 g for 2.5 h to collect large and small vesicles, respectively.

## Western-blot

Extraction of protein from isolated exosomes was performed using SDS2.5%/8 M urea (Sigma-Aldrich) lysis buffer, supplemented with cComplete (Roche) and phenylmethylsulphonyl fluoride (PMSF, Sigma), for 30 min on ice followed by a centrifugation at 17 000 g for 30 min to remove DNA. 30 µg of protein was used for western blot analysis after quantification using DC™ Protein Assay (BIO-RAD). For western-blot analysis of sucrose gradient derived samples, the total volume of 30 µL of each fraction was used. For size exclusion chromatography, 1/3 of the total samples volume was used. Samples were prepared with laemmli buffer and incubated for 10 min at 95 °C, only samples used for anti-reporter or anti-mouse CD63 antibodies incubations were prepared with laemmli buffer without β-mercaptoethanol.

Proteins were run in an SDS-PAGE (sodium dodecyl sulfate-polyacrylamide gel electrophoresis) gel. The Precision Plus Protein™ Dual Color Standards (BIO-RAD) was used as ladder control. After separation by electrophoresis, proteins were transferred onto nitrocellulose membranes 0.2µm (GE Healthcare) using a wet electrophoretic transfer system. Ponceau S staining was used to confirm an effective and equilibrated protein transfer. Subsequently, the nitrocellulose membranes were blocked with 5% non-fat dry milk in PBS 1X/0.1% Tween 20 (Sigma-Aldrich) for 1 h at RT. After blocking,

membranes were incubated overnight at 4 °C on a shaker with the following primary antibodies: anti-mCherry (1:500, Biorbyt orb116118), anti-eGFP (1:500, abcam ab13970), anti-phiYFP (1:1000, Evrogen AB603), anti-mTFP (1:500, Cai Laboratory, Michigan, USA), anti-Syntenin (1:500, abcam ab19903), anti-Alix (Thermo Fisher Scientific MA1-83977), anti-Apolipoprotein A1 (1:500, Novusbio NB600-609), anti-Cytochrome c (1:200, Santa Cruz sc-7159), anti-human CD63 (1:500, BD Pharmingen 556019), and anti-mouse CD63 (1:500, BD Pharmingen 564221). After 4 washes with PBS 1X/ 0.1% Tween 20, membranes were incubated 1 h at RT with horseradish peroxidase (HRP)-conjugated secondary antibodies: anti-goat (1:5000, Abcam ab6741), anti-chicken (1:5000, Sigma-Aldrich A906), anti-rabbit (1:5000, Cell Signalling 7074), anti-rat (1:5000, GenScript a00167), anti-mouse (1:5000, Advansta R-05071-500). Membranes were washed with PBS 1X/0.1% Tween 20 and incubated with Clarity™ Western Enhanced chemiluminescence (ECL) Substrate (BIO-RAD), according to the manufacturer's recommendations, to detect the antibody-specific signal using GE Healthcare Amersham™ Hyperfilm™ ECL. Full uncropped scans can be found in Source Data file.

### Transmission electron microscopy

Samples (5 μl) were mounted on a formvar/carbon-coated 300 mesh nickel grids and incubated for 2 min at RT. After removing the excess liquid with a filter paper, 5 μl of 1% uranyl acetate was added to the grids, left standing for 10 s, and then removed with a filter paper. Imaging was done using a JEOL JEM 1400 TEM at 120 kV (Tokyo, Japan) with a CCD digital camera Orious 1100 W Tokyo, Japan. The transmission electron microscopy was performed at the HEMS core facility at i3S, University of Porto, Portugal with the assistance of Sofia Pacheco and Rui Fernandes.

### Nanoscale liquid chromatography coupled to tandem mass spectrometry (nanoLC–MS/MS)

Three biological replicates were used for WT pancreas small EVs and for KPC small EVs. Each sample was processed for proteomic analysis following the solid-phase-enhanced sample-preparation (SP3) protocol and enzymatically digested with Trypsin/LysC as previously described[54].

Protein identification and quantitation was performed by nanoLC-MS/MS equipped with a Field Asymmetric Ion Mobility Spectrometry–FAIMS interface. This equipment is composed of a Vanquish Neo liquid chromatography system coupled to an Eclipse Tribrid Quadrupole, Orbitrap, Ion Trap mass spectrometer (Thermo Fisher Scientific, San Jose, CA). 250 nanograms of peptides of each sample were loaded onto a trapping cartridge (PepMap Neo C18, 300 μm x 5 mm i.d., 174500, Thermo Fisher Scientific, Bremen, Germany). Next, the trap column was switched in-line to a μPAC Neo 50 cm column (COL-nano050-NeoB) coupled to an EASY-Spray nano flow emitter with 10 μm i.d. (ES993, Thermo Fisher Scientific, Bremen, Germany). A 130 min separation was achieved by mixing A: 0.1% FA and B: 80% ACN, 0.1% FA with the following gradient at a flow of 750 μL/min: 0.1 min (1% B to 4% B) and 1.9 min (4% B to 7% B). Next, the flow was reduced to 250 μL/min with the following gradient: 0.1 min (7.0 to 7.1% B), 80 min (7.1% B to 22.5% B), 30 min (22.5% B to 40% B), 8 min (40%B to 99% B) and 9.9 min at 99% B. Subsequently, the column was equilibrated with 1% B. Data acquisition was controlled by Xcalibur 4.6 and Tune 4.0.4091 software (Thermo Fisher Scientific, Bremen, Germany).

MS results were obtained following a Data Dependent Acquisition –DDA procedure. MS acquisition was performed with the Orbitrap detector at 120 000 resolution in positive mode, quadrupole isolation, scan range (m/z) 375–1500, RF Lens 30%, standard AGC target, maximum injection time was set to auto, 1 microscan, data type profile and without source fragmentation. FAIMS mode: standard resolution, total carrier gas flow: static 4 L/min, FAIMS CV: −45, −60 and −75 (cycle time, 1 s). Internal Mass calibration: Run-Start Easy-IC. Filters: MIPS,

monoisotopic peak determination: peptide, charge state: 2–7, dynamic exclusion 30 s, intensity threshold, 5.0e3. MS/MS data acquisition parameters: quadrupole isolation window 1.8 (m/z), activation type: HCD (30% CE), detector: ion trap, IT scan rate: rapid, mass range: normal, scan range mode: auto, normalized AGC target 100%, maximum injection time: 35 ms, data type centroid.

The raw data was processed using the Proteome Discoverer 3.0.1.27 software (Thermo Fisher Scientific) and searched against the UniProt database for the Mus musculus reviewed Proteome (2023_03 with 17,162 entries). A common protein contaminant list from MaxQuant was also included in the analysis. The Sequest HT search engine was used to identify tryptic peptides. The ion mass tolerance was 10 ppm for precursor ions and 0.5 Da for fragment ions. The maximum allowed missing cleavage sites was set to two. Cysteine carbamidomethylation was defined as constant modification. Methionine oxidation, deamidation of glutamine and asparagine, peptide terminus glutamine to pyroglutamate, and protein N-terminus acetylation, Met-loss, and Met-loss+acetyl were defined as variable modifications. Peptide confidence was set to high. The processing node Percolator was enabled with the following settings: maximum delta Cn 0.05; target FDR (strict) was set to 0.01 and target FDR (relaxed) was set to 0.05, validation based on q-value. Protein label-free quantitation was performed with the Minora feature detector node at the processing step. Precursor ions quantification was performed at the consensus step with the following parameters: unique plus razor peptides were considered, precursor abundance based on intensity, and normalization based on total peptide amount. For hypothesis testing, protein ratio calculation was pairwise ratio-based and an t-test (background based) hypothesis test was performed.

### Cell lines exposure to cancer exosomes

$1 \times 10^6$ CAFs or $8 \times 10^5$ bEnd.3 cells were seeded and, in the following day, each cell line was treated with $5 \times 10^{10}$ (100 μl PBS1x) of KPF CD63-phiYFP exosomes isolated by ultracentrifugation and quantified by NTA as previously described. Exosomes were added to the cells in culture and provided once again 12 h later. Cells were harvested for RNA extraction 24 h post first treatment.

### RNA extraction

RNA from cell samples was obtained following the manufacturer's instructions of the AllPrep DNA/RNA Micro Kit (QIAGEN Cat# 80284). RNA samples were treated with DNase I according to the manufacturer's instructions (Thermo Fisher Scientific EN0521)

For RNA seq analysis of EVs, the pellet of EVs was resuspended in 1 mL of TRItidy G (VWR A4051.0100) and RNA extraction was performed according to manual instructions. Three biological replicates were used for WT pancreas small EVs, three replicates (small EVs isolation from three different tumor pieces) from a KPC mouse and three biological replicates from exosomes collection by ultracentrifugation from a KPC cell line established ex vivo.

### RNA-Seq

RNA concentration and integrity were obtained using Qubit 3.0 fluorometer and Agilent 2100 Bioanalyzer, respectively. Briefly, 10 ng of total RNA was reverse transcribed and the resulting cDNA was amplified for 12 cycles (cell samples) or 16 cycles (EVs samples) by adding PCR Master Mix and the AmpliSeq mouse transcriptome gene expression primer pool (targeting 20 767 well-annotated RefSeq genes + 3 163 XM and XR genes, based on UCSC mm10). Amplicons were digested with the proprietary FuPa enzyme and ligated onto barcoded adapters. The library amplicons were bound to magnetic beads. Libraries were amplified, re-purified and individually quantified using Agilent TapeStation High Sensitivity tape. Individual libraries were diluted to a final concentration of 80 pM and pooled equally for each group of samples for further processing. Emulsion PCR, templating

and 540 chip loading was performed with an Ion Chef Instrument (Thermo Fisher Scientific). Sequencing was performed on an Ion S5XL™ sequencer (Thermo Fisher Scientific).

## Immunofluorescence

**Cell lines.** BxPC-3 cells were plated in coverslips and 24 h after medium was removed and cells were washed with cold PBS 1X followed by fixation with 4% paraformaldehyde (PFA) (Sigma-Aldrich) for 15 min at RT. Coverslips were rinsed three times with PBS1X and incubated with a quenching solution of 0.1 M glycine for 5 min at T. The cells were permeabilized with a solution of Triton-X (VWR) 0.1% followed by a 45 min incubation at RT with 10% Albumin Bovine Fraction V (BSA, NZYTech). After blocking, cells were incubated overnight at 4 °C with the following primary antibodies: anti-human CD63 (Novus Biologicals® H5C6), anti-eGFP (BIO-RAD 81/4745-1051), anti-mCherry, anti-phiYFP and anti-mTFP, in a dilution of 1:500 in PBS 1X/ 2% BSA. Anti-mCherry, anti-phiYFP and anti-mTFP antibodies were developed and kindly provided by Cai Laboratory, University of Michigan Medical School, Michigan, USA. Next day, after 4 washes with PBS 1X, cells were incubated 45 min at RT with the respective secondary antibodies: anti-mouse Alexa-Fluor® 594 (abcam, ab150108), anti-chicken Alexa-Fluor® 633 (Sigma, SAB4600127), anti-sheep Alexa-Fluor® 488 (Jackson ImmunoResearch, 713-545-003), anti-rabbit Alexa-Fluor® 488 (Jackson ImmunoResearch, 711-545-152) and anti-rat Alexa-Fluor® 488 (Invitrogen, A21208), at a 1:500 dilution in PBS 1X/2% BSA. Nuclei counterstain was achieved using hoechst solution (1:10 000, Thermo Fisher Scientific) for 10 min at RT. The coverslips were mounted in glass slides using a drop of VECTASHIELD mounting medium (Vector Laboratories) and sealed with nail polish.

**Formalin-fixed organs.** Tissue samples were fixed in 10% formalin for at least 24 h prior to paraffin embedding. Next, 6μm thick sections were cut using a Microm HM335E microtome, transferred to coated slides (Thermo Fisher Scientific), and left overnight at 37 °C. Sections were deparaffinized and hydrated prior to heat-mediated antigen retrieval with sodium citrate buffer pH 6.0 (Vector Laboratories) for 40 min inside a steamer machine. Afterwards, incubation with a 0.3% hydrogen peroxide in methanol solution ($H_2O_2$ [Sigma-Aldrich]; $CH_3OH$ [VWR]) for 15 min at RT was performed. After washes with PBS 1 X 0.1% Tween 20 tissue sections were bordered with a hydrophobic pen (Vector Laboratories), placed in a humid chamber and incubated with Ultravision Protein-block solution (DAKO) for 1 h at RT. The incubation with the primary antibodies was performed at 4 °C for 1 day in PBS 1 X /0.2% Tween 20. After, slides were washed five times with PBS 1 X /0.1% Tween 20 for 20 min followed by the secondary antibody overnight incubation at 4 °C in a humid chamber.

In Panc-CD63-mCherry and Panc-ExoBow were used the anti-GFP (1:300, BIO-RAD 4745-1051), anti-mCherry (1:500, abcam ab205402), anti-phiYFP (1:300, evrogen #AB603), anti-mTFP (1:500, kindly provided by Cai Laboratory, University of Michigan Medical School, Michigan, USA), anti-CD31 (1:50, abcam ab28364), anti-Alix (1:200, Thermo Fisher Scientific MA1-83977), anti-Syntenin (1:300, abcam ab19903), and anti-Rab7 (1:50, abcam ab137029).

In KPF-CD63-mCherry samples were used the rabbit anti-mCherry 1:500 (kindly provided by Cai Laboratory, University of Michigan Medical School, Michigan, USA), chicken anti-mCherry (1:500, abcam ab205402), anti-αSMA-FITC (1:300, Sigma-Aldrich a2547), anti-CD31 (1:50, abcam ab28364), anti-CD161 (1:1000, abcam ab234107), and anti-Megalin (1:500, abcam ab76969), anti-Aquaporin-2 (1:700, abcam ab199975), anti-Podoplanin (1:100, abcam ab256559), anti-Uteroglobin (1:2000, abcam ab213203), anti-TTF1 (1:200, abcam ab76013).

In KPC-ExoBow tumors were used the anti-mTFP 1:500 (Cai Laboratory, Michigan, USA), anti-CD4 (1:400, abcam ab183685), anti-CD8 (1:300, abcam ab209775), anti-Foxp3 (1:300, abcam ab215206), and anti-CD68 (1:50, abcam ab31630).

In KPC tumors were used the anti-Rab27a (1:50 abcam, ab55667), anti-EpCam (1:500, abcam ab71916) and anti-α-SMA-FITC (1:300 Sigma-Aldrich, a2547).

The secondary antibodies used in samples were anti-rabbit Alexa-Fluor® 488 (Jackson ImmunoResearch JK711545152), anti-rabbit Alexa-Fluor® 546 (Thermo Fisher Scientific, A10040), anti-rabbit Alexa-Fluor® 594 (abcam ab150084), anti-mouse Alexa-Fluor® 488 (abcam ab150105), anti-mouse Alexa-Fluor® 647 (Thermo Fisher Scientific A21235), anti-rat Alexa-Fluor® 488 (abcam ab150153), anti-rat Alexa-Fluor® 594 (Thermo Fisher Scientific A21209), anti-chicken Alexa-Fluor® 633 (Sigma-Aldrich SAB4600127), and anti-sheep Alexa-Fluor® 488 (Jackson ImmunoResearch, 713-545-003).

After incubations, the slides were thoroughly washed with PBS 1 X /0.1% Tween 20. The nuclei were counterstained with hoechst solution (1:10 000, Thermo Fisher Scientific) for 15 min at RT. The slides were washed and mounted with VECTASHIELD mounting medium (Vector Laboratories) and sealed with nail polish.

**Paraformaldehyde-fixed organs.** Mice were anesthetized with ketamine and medetomidine and transcardially perfused using ice-cold PBS 1X followed by ice-cold 4% paraformaldehyde (PFA) in PBS 1X. Pancreas were collected and fixed overnight in 4% PFA, rinsed and incubated overnight in 30% (weight/volume) sucrose solution. After that, organs were embedded in O.C.T. and frozen using dry ice. Frozen sections of about 15μm were made using the Cryostat Leica CM 3050 S (Leica Biosystems). Sections were dried for 1 h at RT followed by washes with PBS 1X. Sodium citrate-based antigen retrieval solution pH 8.5 was pre-heated in a water bath at 80 °C, and slides incubated for 30 min. Slides were rinsed using PBS 1X/Triton-X 0.1% and incubated overnight at 37 °C in PBS 1X/Triton-X 1% with gentle shaking for permeabilization, followed by 10 h incubation in blocking solution (PBS 1X/10% BSA) supplemented with 1% Triton-X at 4 °C with gentle shaking. Next, tissue sections were delimited with a hydrophobic pen (Vector Laboratories), placed in a humid chamber and incubated at 4 °C for one and a half days with the primary antibodies prepared in PBS 1X/Triton-X 0.5%: anti-GFP (1:300, BIO-RAD 4745-1051), anti-mCherry 1:500, anti-phiYFP 1:300, anti-mTFP 1:500. Anti-reporter antibodies were developed and kindly provided by Cai Laboratory, University of Michigan Medical School, Michigan, USA. After incubation with primary antibodies, slides were thoroughly washed using PBS 1X/Triton-X 0.1% and incubated overnight at 4 °C with the respective antibodies anti-mouse Alexa-Fluor® 594 (abcam, ab150108), anti-sheep Alexa-Fluor® 488 (Jackson ImmunoResearch, 713-545-003), anti-rabbit Alexa-Fluor® 488 (Jackson ImmunoResearch JK711545152), anti-rabbit Alexa-Fluor® 546 (Thermo Fisher Scientific, A10040) and anti-rat Alexa-Fluor® 594 (Thermo Fisher Scientific, A21209), in a dilution 1:500. Slides were thoroughly washed and the nuclei counterstained with hoechst solution (1:10 000, Thermo Fisher Scientific) for 15 min at RT. The slides were washed and mounted with VECTASHIELD mounting medium (Vector Laboratories) and sealed with nail polish.

**Microscopy Imaging and analysis.** A Leica TCS SP5 inverted confocal system was used to image samples with an 40x/ N.A. 1.3 or a HCL PL APO CS 63 x / N.A. 1.4 oil objective. Sequential acquisition was used to acquire each fluorescent signal apart from 405 laser for hoescht detection. An upright Zeiss LSM 780 was also used in Fig. 1F, with 40 × 1.3NA oil immersion objective. Presented images were max z-projected (when referenced in Fig. legend), cropped and contrast was optimized in FIJI.

Representative images with z-stacks ($n = 4$ or $n = 3$ in upper and lower examples in Fig. 2 d, respectively) were acquired for each KPC tumor. Quantification was performed in z-maximum projection images obtained in FIJI. Manual ROIs using polygon selection tool for individual PDAC lesions were made based on EpCAM expression in which Rab27a mean fluorescence intensity values were measured. The median value of Rab27a expression in all selected lesions was used as a

threshold to determine high and low Rab27a PDAC lesions. A radial profile with 150 pixels centered in circle that bounds each PDAC lesion ROI was created using the plug-in Radial Profile in Fiji to measure the integrated fluorescence intensity of αSMA surrounding each PDAC lesion. The median radius fitted to the manual PDAC lesions' ROI was on average 59 or 55 pixels for each of the provided examples.

CAFs or bEnd.3 cells were co-cultured with KPF CD63-phyYFP cells (ratio 1:1) in an 8-well high-chamber slide with a tissue culture-treated #1.5 polymer coverslip (Ibidi) in RPMI 1640 Medium, without phenol red (Gibco) supplemented accordingly to each cell line specificities. Imaging was performed at 37 °C and 5% CO2 using a CFI Plan Apo VC 60XC WI /1.2 objective on a Nikon ECLIPSE Ti2 microscope equipped with a CrestOptics X-Light V3 spinning disk and a Photometrics sCMOS Kinetix camera. YFP was excited with a 518 nm laser line and detected through a 560/25 nm emission filter. Images were acquired at day three of co-culture.

### Immunohistochemistry
Samples were fixed in 10% formalin and paraffin embedded. Tissue sections were obtained using a Microm HM335E microtome (Thermo Fisher Scientific) and the immunohistochemistry was performed as previously described[30]. Heat-mediated antigen retrieval for 40 min using TRIS-EDTA pH 9 solution was used for CD31 (1:50, abcam ab28364). Quantification of CD31+ vessels was performed in at least 5 random fields (10x).

### Statistical analysis
Significance between conditions was determined at *$p$ < 0.05, **$p$ < 0.01, ***$p$ < 0.001 and ****$p$ < 0.0001. All analyses were performed in GraphPad Prism®.

### Reporting summary
Further information on research design is available in the Nature Portfolio Reporting Summary linked to this article.

## Data availability
Figures data have associated raw data included in Source Data file: Figs. 1–7 and Supplementary Figs. 1-9,11,12. The mass spectrometry proteomics data have been deposited to the ProteomeXchange Consortium via the PRIDE partner repository with the dataset identifier PXD047009. RNA Seq data generated within this study has been submitted to European Nucleotide Archive (ENA) browser under the accession code PRJEB71061. The remaining data are available within the Article, Supplementary Information or Source Data file. Source data are provided with this paper.

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

## Acknowledgements

This work was supported by Portuguese funds through FCT—Fundação para a Ciência e a Tecnologia in the framework of the project 2022.02284.PTDC and by Programa Operacional Regional do Norte co-funded by European Regional Development Fund under the project "The Porto Comprehensive Cancer Center" (DOI 10.54499/PINFRA03/72678/2020), NORTE-01-0145-FEDER-072678—Consórcio PORTO.CCC—Porto. Comprehensive Cancer Center. B.A., N.B., C.F.R and I.A.B. were funded by the Portuguese Foundation for Science and Technology (PD/BD/135546/2018, SFRH/BD/130801/2017, SFRH/BD/131461/2017 and SFRH/BD/144854/2019). The authors acknowledge the support of the i3S Scientific Platforms: Translational Cytometry, Animal Facility, Advance Light Microscopy, Genomics, Proteomics, Bioimaging, and Histology and Electron Microscopy (HEMS). Bioimaging and HEMS are members of the national infrastructure PPBI—Portuguese Platform of Bioimaging (PPBI-POCI-01-0145-FEDER-022122). Mass spectrometry was performed with the assistance of Hugo Osório, that had the support from the Portuguese Mass Spectrometry Network, integrated in the National Roadmap of Research Infrastructures of Strategic Relevance (ROTEIRO/0028/2013; LISBOA-01-0145-FEDER-022125). Schemes were created using BioRender.com.

## Author contributions

Conceptualization, S.A.M, B.A. and C.F.R; Methodology, B.A., D.C. and S.A.M.; Formal analysis, B.A.; Investigation, B.A., N.B., C.F.R., P.F.V., S.S.A., C.D., B.C. and I.A.B.; Resources, D.S., D.C. and S.A.M.; Data curation: B.A.; Writing—original draft, B.A. and S.A.M; Writing—Review & Editing, N.B., J.C.M., and S.A.M; Visualization, B.A.; Supervision, S.A.M. Funding acquisition, S.A.M.

## Competing interests

S.A.M. holds patents in the area of EVs biology (miRNA biogenesis in exosomes for diagnosis and therapy, publication number 20200255831; use of exosomes for the treatment of disease, patent number 10959952; Analysis of genomic DNA, RNA and proteins in exosomes for diagnosis and theranosis, publication number 20200200755). The other authors declare no competing interests.
