## [Peer Review File · Nature Communications]

Exosomes define a local and systemic communication network in healthy pancreas and pancreatic ductal adenocarcinomaREVIEWER COMMENTS

Reviewer #1 (Remarks to the Author):

In this manuscript, Adem et al generated a mouse model that can overexpress CD63 and tag the CD63 with fluorescent proteins in a cell type-specific inducible manner. Generating a tracing system that can keep track of exosome in vivo might be useful for the field, however, the current models were driven by a strong synthetic CAG promoter. Therefore, although CD63 was expressed in a cell specific manner, such expression may not be physiologically relevant. In addition, major results from this manuscript were already reported by many previous studies, for example, cancer cells communicate with stromal cells by exosomes. Another main issue is the lack of detailed characterization. For example, although fluorescent signals were shown in stromal cell or immune cells, however, it is not necessary that these cells were uptaking exosomes. For example, it could be cell debris or non-exosome vesicles. Below are more detail comments:

1. Fig 1f, eGFP and mTFP looked like there's membranous signal, while there's no such signal in phiYFP. It probably suggests these fluorescent proteins might differentially affect the trafficking and potentially function of CD63.
2. Fig 1d, why the molecular weight of CD63-mCherry is different from Sup Fig 2f?
3. As one of the models, Ad-CMV-Flpo was injected orthotopically to induce the allele recombination. However, such Flpo is not cell type specific and may cause CD63 expression in other cell types.
4. It was assumed that "No major differences were observed between the different reporter models, hereinafter all referred to as the KPC-ExoBow, concerning pancreas tumor histology or disease progression (Supplementary Fig. 5b)." This was not true, since inducible Cre is different from constitutively active Cre, in which constitutively active Cre will be active during development and results in a broader distribution of recombination in the pancreas.
5. Line 207, "Considering the prognostic significance of α -smooth muscle actin-positive CAFs in PDAC progression much due to their association with an inflammatory phenotype". This is not true, myCAF is not associated with an inflammatory phenotype (PMID: 28232471).
6. Phenotypes of stromal cells uptaking exosomes need to be characterized better. For example, what phenotypic changes it leads to? This can be done by sorting out different populations of CAFs or endothelial cells that have CD63 signals and compare their signatures to the ones that are CD63 negative. Or CD63+ exosomes can be enriched from the tumor to treat stromal cells to study the phenotypic changes.
7. Fig 5f, lung signal was very modest. In addition, what cell types were uptaking exosomes in each organ needs to be explored. Were major cells in these organs really uptaking or this is because these organs just have more blood supply with more exosomes circulating?

Reviewer #2 (Remarks to the Author):

The manuscript, exosomes define a local and systemic communication network in healthy pancreas and in pancreatic ductal adenocarcinoma is well written and clearly structured with natural progression and assessment of the GEMM (Exbow) model to trace exosomes from spontaneous mouse disease models of PDAC and healthy pancreas.

This mouse is an exciting tool for the field and provides a model system to track exosome in vivo. Its future application for other disease models or cancer type is touched upon in the discussion but should be more broadly described for future use in the field. The caveats of a CD63 marker only approach- is also discussed and correctly point out that no 'one model fits all approach' will be full proof and therefore the authors correctly point out its applicability and limitations for tracking but also its benefits in this context also. While this is true and discussed the authors should soften their statements throughout the MS of hard statements such as - no pre metastatic niche exosome or effect on the liver are found- as other more subtle approaches to exosome tracing could/have shown this using other markers – a simple statement pointing out that other marker may show this in PDAC or in diff disease states and setting will suffice.

Reviewer #3 (Remarks to the Author):

In the manuscript "Exosomes define a local and systemic communication network in healthy pancreas and in pancreatic ductal adenocarcinoma" by Adem et al., the authors generate different genetically modified models of fluorescence labelled CD63 in order to trace exosome uptake in healthy pancreas and pancreatic cancer mouse models. Overall, the generation of the models is highly interesting and will provide a relevant platform technology for extracellular vesicle research. Most of the experiments seem to have been conducted appropriately, however, the data is partly over-interpreted, some claims and hypothesis are unfounded and a couple of key observations are missing.

First of all, the authors use the phrase 'communication' very loosely. As outlined below, I do not think that any level of 'communication' has been established. The 'communication network' in the title is misleading at best. Generally, the title should be changed to more closely reflect the data, which is the generation and testing of a fluorescent CD63 mouse model in healthy pancreas and pancreatic cancer.

The authors might consider the following points to improve the quality and clarity of the manuscript:

Line 63: The exosome reference from 2013 (ref4) needs to be updated. It is highly questionable that, according to the misev2018 guidelines, there is sufficient evidence for calling the vesicles here exosomes. Please add a statement why you are using the 'exosome' label and not small extracellular vesicles.

-Sfig 1b-d, GFP-CD63 cell line, please provide data demonstrating the percentage of GFP positive cells in the CD63-GFP line, and that the exosomes are GFP positive.

-Sfig 2: Similar, can the percentage of positive cells and exosomes be quantified. For 2d, f; please provide the same staining/protein evaluation for untransfected cells as control (including the XFP staining). The authors state that the isolation of exosomes was done according to misev2018, could the authors please also demonstrate (including but not restricted to Sfig2, Fig1) that the isolated fractions are having the features requested as minimal evidence in misev2018 for exosomes/sEV?

L154ff, Fig1d: "We found CD63-mCherry to be enriched in the small EVs fraction, which indicates a clear enrichment in exosomes (Fig. 1d)." Could the authors please provide a reason for this strong statement? It is often described that large EVs are also CD63 positive, and CD63 is not an exosome marker. Also, it is noted that the authors claim to separate the sEV and lEV by differential centrifugation speeds. Please provide characterization of the sEV and lEV in Fig 1d and 1g demonstrating that these are non-overlapping populations. Provide a size distribution. Add information how many particles and protein was loaded in each lane of 1d, g. Given the massive variations of mCherry in M1 and M2 in both sEV and lEV in 1d, is n=2 sufficient for a strong statement as done here in results?

Fig1: What is the age of the animals in the respective panels?

Line 167: "which reflects either incomplete Cre" half sentence with 'OR' is missing.

Line188ff: While it is good to see that no differences between the Flp approaches were observed, please clearly mark in the figure legends and results which specific approach was used in each panel to ensure data can be reproduced more easily.

Line200: Fig2a shows the uptake of two source exosome mice, one n=2 and one n=3. Please distinguish the sources/uptake by using different symbols or colors in 2a.

Line200: A general, big problem of the paper is the concept that uptake from here equalled to 'communication' (line 200). The data only demonstrates uptake, but not whether any of the exosomes have any functional impact. This would be the basic requirement for a first step in a (one-way only) communication. Please define the phenotypical/functional impact of PDAC exosomes on CAFs, endothelial or immune cells.

A key suggestion of the data is that there is uptake and suggested functional/phenotypic impact of pancreatic or PDAC exosomes on stromal cells. Is there a difference in the 'communication' and the resulting phenotypes of stromal cells between pancreatic and PDAC exosomes?

Line229-231: "Collectively, this suggests that modulation of angiogenesis could be more dependent on the cargo of exosomes rather than quantity/frequency of communication." This conclusion is questionable as no data is provided to substantiate it. Is there a difference in cargo? Is there a difference in quantity or frequency of exosomes being secreted? While a comprehensive, multiomics analysis of PDAC and healthy exosome cargo would be suitable, the very least would be a thorough evaluation of the frequency of exosomes (healthy/PDAC) in blood/lymph, and correlate those with the occurrences of uptake. See also comment below, Fig5.

Fig3d-g: While this approach is showing the difference between healthy pancreas and PDAC exosomes on angiogenesis, should the PDAC model not also be performed in a healthy pancreas Rab27ako background? Would PDAC exosomes not also be taken up by healthy pancreas and modulate their (angiogenesis altering) phenotype?

Similar to above, Fig4 is difficult to interpret as it is not shown if there is a difference locally or systemically between healthy pancreas and PDAC exosomes. The increased uptake (not communication) in immune cells could be attributed to a difference in abundance of exosomes.

Fig4c and 4d: In 4c, about 8% of T cells are CD63XFP positive, ie took up exosomes. How is it possible that in 4d, each to Th and Tc are about 6% positive, in addition to a small Treg percentage? Should Th + Tc + Treg not be t cells from 4c?

Fig5 and above: The authors achieved to generate a traceable, PDAC-specific exosome model. A key question in the cancer exosome field is the relative amount of cancer exosomes in body fluids, for pancreatic cancer maybe best blood. Using their ExoBow model, could the authors determine the abundance and clonal heterogeneity of cancer/PDAC exosomes in the blood? Or with the CD63-mCherry at least the abundance in the blood?

This would be very informative to support data from Fig5, which while interesting in itself, should be strengthened by exploring the 'blood' and/or 'lymph' intermediate exosome abundance.

SFig7c,d: as above, is this observation a result of less uptake in the organs for healthy pancreas, or a result of reduced healthy pancreas exosome in circulation? Could the above suggested evaluation of blood circulating PDAC exosomes be expanded for healthy pancreas exosomes? The conclusion currently that the increased uptake is due to unexplored content differences is rather far fetched.

The discussion should be revisited. In general, a restatement of the achievements of the paper should be avoided. The speculation of line340-349 needs to be made more precise. The suggestion of antigen presentation (line358) is not at all addressed in the study and the rather rudimentary immunological data does not allow any such suggestion.

The discussion could focus on applications of the ExoBow model beyond PDAC/pancreatic cancer or cancer in general. What other applications could be found for such a model? What is the current state of 'tracing exosomes' and which problems can be overcome with the ExoBow model? A key limitation is the linkage to CD63 as THE exosome marker. However, several previous publications, including PMID: 26858453 (your reference 13), 34282141, 30949309, show that 'exosomes' are not all CD63 positive.

Furthermore, an honest reflection and distinction from other fluorophore or luminescence tracing approaches (for example PMID: 33042758) should be discussed and evaluated.

Minor point:

Please correct the figure legend to 4 from NK to NK1.1

Reviewer #1 (Remarks to the Author):

In this manuscript, Adem et al generated a mouse model that can overexpress CD63 and tag the CD63 with fluorescent proteins in a cell type-specific inducible manner. Generating a tracing system that can keep track of exosome in vivo might be useful for the field however, the current models were driven by a strong synthetic CAG promoter. Therefore, although CD63 was expressed in a cell specific manner, such expression may not be physiologically relevant.

R: We appreciate the reviewer's valuable feedback, which has significantly improved the quality of our manuscript.

CD63 serves as a widely recognized exosomal marker (Baietti, Zhang et al. 2012, Kowal, Arras et al. 2016) and in our study functions exclusively as a tracking tool. We have characterized small and large EVs isolated directly from the pancreas of *wild-type*, CD63-mCherry, and ExoBow mice. Our assessment included nanoparticle tracking analysis (NTA; Supplementary Fig. 7g) and transmission electron microscopy, revealing similar size and morphology between the distinct models (Supplementary Fig. 7f). Additionally, we demonstrate co-localization of CD63-XFP proteins with key exosomes biogenesis markers, including Alix, syntenin, and Rab7 (Baietti, Zhang et al. 2012, Addi, Presle et al. 2020). While the co-localization with Rab7 is less pronounced due to its association with multivesicular bodies directed to lysosomal compartments (Supplementary Fig. 5b) (Langemeyer, Fröhlich et al. 2018), our collective evidence strongly supports the endocytic nature of CD63-XFP vesicles. Notably, CD63 overexpression does not influence disease progression in PDAC mice (Supplementary Fig. 1b-e), confirming that CD63-XFP expression does not yield non-physiological outcomes.

The decision to insert the transgene into the ROSA-26 locus, rather than the CD63 locus, offers several advantages: 1) it avoids disrupting the expression of the endogenous gene; 2) it maintains stable reporter expression over time; and 3) it enhances the number of CD63-XFP molecules per vesicle, thereby improving the signal-to-noise ratio and the sensitivity for identifying EV-positive cells (Dou, Lin et al. 2021, Xie, Wang et al. 2022). Importantly, this strategy has been successfully employed in other models for EV tracing (Yoshimura, Kawamata et al. 2016, Men, Yelick et al. 2019, Neckles, Morton et al. 2019, McCann, Bischoff et al. 2020, Li, Wang et al. 2022, Nørgård, Steffensen et al. 2022).

Further discussion of this topic can be found in our manuscript (page 16).

In addition, major results from this manuscript were already reported by many previous studies, for example, cancer cells communicate with stromal cells by exosomes.

R: To the best of our knowledge, our study represents the first instance of utilizing an exosomes tracking allele to investigate the natural trafficking of exosomes originating from both healthy pancreas and PDAC tissues in GEMMs. Our work offers several pioneering contributions, including:

- 1. Demonstrating that the healthy pancreas communicates via exosomes with endothelial cells and immune cells in the pancreas (Fig. 3 and Supplementary Fig. 9c).**

2. Revealing that the healthy pancreas communicates via exosomes with thymus, brain, intestine, and bone-marrow (Supplementary Fig. 10c,d).
3. Showing that PDAC cells engage in exosomes-based communication with CAFs, endothelial cells, and immune cells within the tumor microenvironment (Fig. 2, 3 and 4).
4. Identifying the cell types which uptake PDAC exosomes in the most frequent sites of communication, the kidneys and lungs. In particular we observed megalin and aquaporin-2 positive cells in the kidneys and uteroglobin and podoplanin positive cells in the lungs (Fig. 6).
5. Addressing the absence of compelling evidence supporting the formation of a pre-metastatic niche through the direct uptake of exosomes by cells at the liver (Fig. 5b-d).

It is important to note that all of these findings are novel and distinct from previous models and observations, as earlier approaches primarily relied on cell lines injected into mice and lacked the spontaneously exosomes secretion observed in our GEMM-based study.

Another main issue is the lack of detailed characterization. For example, although fluorescent signals were shown in stromal cell or immune cells, however, it is not necessary that these cells were uptaking exosomes. For example, it could be cell debris or non-exosome vesicles.

R: While we acknowledge the possibility of occasional phagocytic events involving cellular debris, our data indicates that cells with prominent phagocytic characteristics within the tumor microenvironment do not exhibit considerable CD63-XFP signal (Fig. 2a, 4c and Supplementary Fig. 9c). This observation strongly suggests that phagocytosis of cellular debris is not the primary source of the observed signals.

Below are more detail comments:

1. Fig 1f, eGFP and mTFP looked like there's membranous signal, while there's no such signal in phiYFP. It probably suggests these fluorescent proteins might differentially affect the trafficking and potentially function of CD63.

R: We appreciate the reviewer's point. Although the fluorescent signal pattern is endosomal for the four colors, we attribute the variations in membrane localization to differences in expression levels. Nonetheless, our findings confirm that the pancreas of Panc-CD63-mCherry and Panc-ExoBow mice releases EVs that are positive for all four fluorescent markers (Fig. 1g and Supplementary Fig. 6c). This underscores that CD63-tagged EVs are secreted into the extracellular space independently of the reporter.

To further support the consistency of our observations, we cloned identical sequences resulting from Flp and Cre recombination of the ExoBow transgene and transfected them into a PDAC cell line. Our results reveal that the subcellular localization of the four CD63-XFP proteins is indistinguishable, and they co-localize with the endogenous CD63 protein (Supplementary Fig 2c-e). Notably, the number of secreted exosomes from each CD63-XFP cell type is not significantly

different, with each secreting exosomes marked with their respective colors (Supplementary Fig 4a-d). Therefore, there are no significant differences in expression patterns or secretion between color-coded exosomes.

Moreover, we established monocolored PDAC cell lines from tumors of KPC CD63-mCherry and KPC-ExoBow mice and found that the expression pattern of each CD63 fusion protein is identical (Supplementary Fig. 3). We quantified the number of circulating exosomes in the serum of the models used and found no significant differences between them (Supplementary Fig. 7d). Importantly, CD63-XFP proteins do not influence disease progression (Supplementary Fig. 1b-e).

Most importantly, we demonstrate that regardless of the color analyzed (CD63-XFP), endothelial cells are one of the most consistent targets of communication in the pancreas (Supplementary Fig. 9b). Thus, our evidence strongly supports the notion that all four CD63-XFP proteins exhibit similar behavior.

2. Fig 1d, why the molecular weight of CD63-mCherry is different from Sup Fig 2f?

R: We appreciate the reviewer's observation. You are correct, the molecular weight numbers were not correctly indicated in the figure montage. However, we have included the raw data in the source data file provided with the initial submission (<https://figshare.com/s/bb940a672a5642a0a2e5>), which clearly demonstrate that the molecular weight is indeed consistent across the blots:

Supplementary Figure 4 b

Full blot of sucrose gradient of BxPC-3 CD63-mCherry exosomes

3. As one of the models, Ad-CMV-Flpo was injected orthotopically to induce the allele recombination. However, such Flpo is not cell type specific and may cause CD63 expression in other cell types.

R: The orthotopic injection of Ad-CMV-Flpo resulted in Flp-driven recombination of the ExoBow transgene in various cell types, not limited to pancreatic cancer cells. However, it is important to note that this injection was specifically carried out in KPC-ExoBow Flp negative mice (*LSL-Kras^{G12D/+}; LSL-Trp53^{R172H/+}; Pdx1-Cre; R26^{CD63-XFP/+}*). In these mice, the Pdx1-Cre allele exclusively recombines the ExoBow transgene for CD63-eGFP, -phiYFP or -mTFP within pancreatic cancer cells (Fig. 1a).

Our analysis focuses solely on the fluorescence from CD63-eGFP, -phiYFP or -mTFP positive cells, which are derived exclusively from pancreatic cancer cells. We apologize for any prior lack of clarity on this matter and have now included a schematic to better elucidate this process in the manuscript (Supplementary Fig. 8a).

4. It was assumed that “No major differences were observed between the different reporter models, hereinafter all referred to as the KPC-ExoBow, concerning pancreas tumor histology or disease progression (Supplementary Fig. 5b).” This was not true, since inducible Cre is different from constitutively active Cre, in which constitutively active

Cre will be active during development and results in a broader distribution of recombination in the pancreas.

R: Thank you for your feedback. The sentence in question is accurate. We did not observe significant histological or disease progression differences between the models. We apologize if the H&E photos may have been misleading due to variations in the timing of their collection. We have replaced them with recent H&E images (Supplementary Fig. 8b).

For the histopathological analysis of animals used in the study of PDAC-derived exosomes biodistribution, please refer to Supplementary Fig. 10e. Furthermore, we have included a comparison of tumor volumes in mice euthanized between 19 and 22 weeks, demonstrating no differences between the control group (KPC) and the CD63-XFP models (KPC-ExoBow or KPC-iExoBow).

Regarding the comment on inducible and constitutively active Cre, we apologize if we were not clear in the manuscript. To clarify, we do not have an inducible Cre; instead, we use an inducible Flp (R26-LSL-FlpoERT2). This setup means that the model is inducible only for the expression of CD63-XFP. The Cre recombination takes place at the same time point as in the non-inducible model. The reason CD63-XFP is not expressed initially is because Flp has not yet excised the STOP cassette, but Cre recombination occurred earlier in both cases.

5. Line 207, “Considering the prognostic significance of α -smooth muscle actin-positive CAFs in PDAC progression much due to their association with an inflammatory phenotype”. This is not true, myCAF is not associated with an inflammatory phenotype (PMID: 28232471).

R: We thank the reviewer for raising this valid point. We have revised the manuscript to clarify the message we intended to convey (page 9).

6. Phenotypes of stromal cells uptaking exosomes need to be characterized better. For example, what phenotypic changes it leads to? This can be done by sorting out different populations of CAFs or endothelial cells that have CD63 signals and compare their signatures to the ones that are CD63 negative. Or CD63+ exosomes can be enriched from the tumor to treat stromal cells to study the phenotypic changes.

R: We appreciate the reviewer’s insight. To address this concern, we established a primary KPF CD63-phiYFP cancer cell line derived from a KPF CD63-mCherry tumor, as detailed in the Material and Methods. Subsequently, we isolated exosomes from KPF CD63-phiYFP cancer cells and employed them to treat a CAFs cell line established from a KPC tumor, and a mouse endothelial cell line. Our

analysis confirmed the presence of CD63-phiYFP exosomes in both CAFs and endothelial cells (Supplementary Fig. 11c, d).

To gain deeper insights into the phenotypic changes induced by these exosomes, we conducted RNASeq analysis. This investigation revealed 301 and 292 differentially expressed genes in CAFs and endothelial cells, respectively, following exposure to cancer exosomes (Fig. 7i-k). Notably, in CAFs, the differentially expressed upregulated genes were associated with pathways related to cell differentiation, enhanced cell adhesion, reduced cell proliferation, programmed cell death, and alterations in protein metabolism. These findings align with the concept of metabolic reprogramming in CAFs, which can influence cancer cell metabolic switch and growth capacity (Li, Sun et al. 2021). Furthermore, we observed gene pathways related to immune response regulation, which may be linked to the recently described antigen-presenting CAFs subtype, with its biological impact yet to be fully elucidated.

In endothelial cells, we identified a significant set of downregulated genes related to cell adhesion and differentiation, resulting in a reduction in blood vessel formation, consistent with our earlier *in vivo* observations (Fig. 3).

In summary, our findings demonstrate that both CAFs and endothelial cells undergo gene expression alterations upon exposure to cancer exosomes, aligning with our previous *in vivo* findings (Fig. 2 and 3).

7. Fig 5f, lung signal was very modest. In addition, what cell types were uptaking exosomes in each organ needs to be explored. Were major cells in these organs really uptaking or this is because these organs just have more blood supply with more exosomes circulating?

R: The signal intensity in the lungs is indeed weaker than in the kidneys, as clearly depicted in Fig. 5d and Fig. 5f.

Our data does not support the notion that the presence of CD63-XFP exosomes in specific organs correlates with their vascularity or anatomical proximity to the pancreas. The liver, despite receiving approximately 25% of the cardiac output, does not exhibit significant signal intensity (Fig. 5d), nor does the signal increase from healthy to PDAC states (Fig. 5b). Moreover, the anatomical positioning of the pancreas, portal circulation, and splenic vessels might suggest the liver and spleen as prime candidates for exosomes accumulation, but this is not the case.

To identify the specific recipient cells within the kidneys and lungs, we employed the following markers: in the kidneys, megalin (proximal tubule cells), aquaporin-2 (collecting duct cells) and podoplanin (glomeruli cells); in the lungs, uteroglobin (non-ciliated epithelial Clara cells), podoplanin (type-I pneumocytes) and TTF1 (type II alveolar cells and club cells; Fig. 6).

In the kidneys, the major recipients of PDAC exosomes were proximal tubules, followed by collecting ducts. Notably, glomeruli cells did not exhibit positivity for PDAC exosomes, a finding consistent across both early and late PDAC stages. These observations align with our IVIS analysis, which revealed an increase in PDAC exosomes signal at late-stage disease compared to early stages in the kidneys.

In the lungs, Clara cells and type-I pneumocytes were identified as the recipients of PDAC exosomes, while type II alveolar cells and club cells were not involved. This pattern in the lungs was also consistent with our IVIS data, depicting similar signals in early and late PDAC stages.

Reviewer #2 (Remarks to the Author):

The manuscript, exosomes define a local and systemic communication network in healthy pancreas and in pancreatic ductal adenocarcinoma is well written and clearly structured with natural progression and assessment of the GEMM (Exbow) model to trace exosomes from spontaneous mouse disease models of PDAC and healthy pancreas.

This mouse is an exciting tool for the field and provides a model system to track exosome *in vivo*. Its future application for other disease models or cancer type is touched upon in the discussion but should be more broadly described for future use in the field. The caveats of a CD63 marker only approach- is also discussed and correctly point out that no 'one model fits all approach' will be full proof and therefore the authors correctly point out its applicability and limitations for tracking but also its benefits in this context also. While this is true and discussed the authors should soften their statements throughout the MS of hard statements such as - no pre metastatic niche exosome or effect on the liver are found- as other more subtle approaches to exosome tracing could/have shown this using other markers – a simple statement pointing out that other marker may show this in PDAC or in diff disease states and setting will suffice.

R: We would like to express our gratitude to the reviewer for the positive feedback and valuable constructive input, which has significantly enhanced the quality of our manuscript. In response to your suggestions, we have expanded the discussion on the model's potential for future studies and prospective applications (see page 16).

It is important to clarify that our intent was never to dismiss the possibility of the pre-metastatic niche occurring with alternative markers or in different physiological or pathological contexts. We have refined our statements regarding communication with the liver and emphasized that our observations are limited to CD63⁺ exosomes originating from the pancreas within the context of the described models (please see the discussion on page 13).

Throughout the manuscript, we have taken care to avoid overinterpretation of results, while still allowing for the exploration of exosomes' *in vivo* biological relevance based on our observations and the existing literature.

Reviewer #3 (Remarks to the Author):

In the manuscript “Exosomes define a local and systemic communication network in healthy pancreas and in pancreatic ductal adenocarcinoma” by Adem et al., the authors generate different genetically modified models of fluorescence labelled CD63 in order to trace exosome uptake in healthy pancreas and pancreatic cancer mouse models. Overall, the generation of the models is highly interesting and will provide a relevant platform technology for extracellular vesicle research. Most of the experiments seem to have been conducted appropriately, however, the data is partly over-interpreted, some claims and hypothesis are unfounded and a couple of key observations are missing. First of all, the authors use the phrase ‘communication’ very loosely. As outlined below, I do not think that any level of ‘communication’ has been established. The ‘communication network’ in the title is misleading at best. Generally, the title should be changed to more closely reflect the data, which is the generation and testing of a fluorescent CD63 mouse model in healthy pancreas and pancreatic cancer.

R: We thank you for your positive feedback and for the effort put into providing constructive assessments that have greatly improved our manuscript. We have thoroughly addressed all your comments in a point-by-point response and have also carried out additional experiments as suggested.

The authors might consider the following points to improve the quality and clarity of the manuscript:

Line 63: The exosome reference from 2013 (ref4) needs to be updated. It is highly questionable that, according to the MISEV2018 guidelines, there is sufficient evidence for calling the vesicles here exosomes. Please add a statement why you are using the ‘exosome’ label and not small extracellular vesicles.

R: We have updated reference 4 (Raposo, G. et al. JCB 2013) to (van Niel, D'Angelo et al. 2018).

While we acknowledge the absence of a bona fide marker for exosomes and vesicles in general, we have emphasized this in our manuscript (pages 14 and 15). However, CD63 is consistently recognized in the literature as the most prominent marker associated with exosomes (Kowal, Arras et al. 2016, Zhang, Freitas et al. 2018, Mathieu, Névo et al. 2021).

In our manuscript, we have conducted experiments following MISEV 2018 guidelines to validate that CD63-XFP is enriched in exosomes, which include: 1) vesicle characterization via nanoparticle tracking analysis and electron microscopy (Supplementary Fig. 4a and 7f,g); 2) vesicles isolation by sucrose gradient and evaluation of fractions (Supplementary Fig. 4c); 3) vesicle isolation by size exclusion chromatography (Supplementary Fig. 4e); and 4) characterization of markers such as CD63, syntenin, apolipoprotein-A1, and cytochrome C (Supplementary Fig. 4e,f and 5c,d).

The nomenclature of small extracellular aims to address the challenges in distinguishing exosomes from small ectosomes during *in vitro* isolation and purification processes. However, in our study, we analyze the spontaneous flow of CD63 positive vesicles without isolating EVs for injection into mice. Given the

prevalence of CD63 expression on vesicles of endosomal origin, particularly exosomes, we believe this nomenclature is the most accurate in the specific context of our study.

-Sfig 1b-d, GFP-CD63 cell line, please provide data demonstrating the percentage of GFP positive cells in the CD63-GFP line, and that the exosomes are GFP positive.

R: As stated in the methods section (page 16), the CD63-GFP cell line was previously described and validated by us in PMID: PMC9271144 (Carolina, Nuno et al. 2022). We have now included data demonstrating that the CD63-GFP cell line is 99.5% positive for GFP (Supplementary Fig. 1c). Additionally, we conducted Image Stream flow cytometry to confirm that MIA PaCa-2 CD63-GFP exosomes are fluorescent, although this data is not included in the manuscript as the MIA PaCa-2 CD63-GFP cell line was primarily used to track tumor growth rather than assess exosomes biodistribution.

-Sfig 2: Similar, can the percentage of positive cells and exosomes be quantified. For 2d, f; please provide the same staining/protein evaluation for untransfected cells as control (including the XFP staining). The authors state that the isolation of exosomes was done according to misev2018, could the authors please also demonstrate (including but not restricted to Sfig2, Fig1) that the isolated fractions are having the features requested as minimal evidence in misev2018 for exosomes/sEV?

R: We have included the percentage of positive BxPC3 cells and their respective exosomes in the manuscript (Supplementary Fig. 2b and 4d). Previously, we provided anti-huCD63 staining of untransfected cells in Supplementary Fig. 2c. Additionally, we have now included anti-XFP immunofluorescence of non-transfected cells (scale bar: 20µm).

Furthermore, to ensure the specificity of the reporter protein antibodies, we isolated exosomes from both non-transfected cells and CD63-XFP fluorescent clones and conducted anti-XFP western-blot analysis, which confirms the antibodies' specificity (Supplementary Fig. 4b).

In line with the MISEV 2018 guidelines for exosomes/sEVs, we performed the following procedures:

1. Nanoparticle tracking analysis (supplementary Fig. 4a and 7g).
2. Isolation of vesicles by size exclusion chromatography (SEC, Supplementary Fig. 4e,f).
3. Characterization of EV populations (small vs. large or SEC fractions) by western blot, using CD63 and syntenin as positive markers of exosomes, and ApoA1 and cytochrome C as negative controls (Supplementary Fig. 4e,f and 5c,d).
4. Isolation of both small and large EVs, followed by transmission electron microscopy (Supplementary Fig. 7f).

L154ff, Fig1d: "We found CD63-mCherry to be enriched in the small EVs fraction, which indicates a clear enrichment in exosomes (Fig. 1d)." Could the authors please provide a

reason for this strong statement? It is often described that large EVs are also CD63 positive, and CD63 is not an exosome marker.

R: We appreciate your comment. Indeed, we acknowledge that CD63 is not an exclusive marker of exosomes but is enriched in these vesicles, and this information is clearly stated in the manuscript (pages 3, 7, 15 and 16). Our data indicates that, for the same protein loading, CD63-XFP proteins are enriched in the small EVs fraction, which predominantly contains exosomes, as opposed to the large EVs fraction (Fig. 1d, g and Supplementary Fig. 6c). Moreover, we isolated small and large EVs from *wild-type* mice and confirmed the enrichment of endogenous CD63 protein in the small EVs fraction (Supplementary Fig. 5c). We have also rephrased our statement to address this (page 7).

Also, it is noted that the authors claim to separate the sEV and lEV by differential centrifugation speeds. Please provide characterization of the sEV and lEV in Fig 1d and 1g demonstrating that these are non-overlapping populations. Provide a size distribution. Add information how many particles and protein was loaded in each lane of 1d, g. Given the massive variations of mCherry in M1 and M2 in both sEV and lEV in 1d, is n=2 sufficient for a strong statement as done here in results?

R: We have characterized the small and large EVs populations through nanoparticle tracking analysis (Supplementary Fig. 7g), western-blot for CD63, syntenin, Alix, ApoA1, and cytochrome C (Supplementary Fig. 5c,d), as well as electron microscopy (Supplementary Fig. 7f).

The protein load for Fig. 1d and 1g has been included in figure legend 1.

While there are variations between mouse 1 and mouse 2 in the western-blot of Fig. 1d, it is evident that the small EVs fraction consistently exhibits enrichment in comparison to the large EVs fraction in both mice (Fig. 1d). Furthermore, we have isolated small and large EVs from a third mouse and have included the western-blot for mCherry, reaffirming our findings that CD63-mCherry is enriched in the smaller EVs population.

Fig1: What is the age of the animals in the respective panels?

R: We have now included the ages of the animals in the figure legend for clarification.

Mice age:

- Panels b, c, e and f is 8 weeks.
- Panels d and g is between 8 to 11 weeks.
- Panels h and i is 16.3 weeks.
- Panels j and k is 17 weeks.

Line 167: “which reflects either incomplete Cre” half sentence with ‘OR’ is missing.

R: Thank you. The sentence was corrected (page 8).

Line188ff: While it is good to see that no differences between the Flp approaches were observed, please clearly mark in the Fig. legends and results which specific approach was used in each panel to ensure data can be reproduced more easily.

R: Done.

Line200: Fig2a shows the uptake of two source exosome mice, one n=2 and one n=3. Please distinguish the sources/uptake by using different symbols or colors in 2a.

R: Done.

Line200: A general, big problem of the paper is the concept that uptake from here equalled to ‘communication’ (line 200). The data only demonstrates uptake, but not whether any of the exosomes have any functional impact. This would be the basic requirement for a first step in a (one-way only) communication. Please define the phenotypical/functional impact of PDAC exosomes on CAFs, endothelial or immune cells.

R: Thank you for your comment. In our study, we use the term “communication” in its strict sense, referring to the process by which information is sent from one place to another. We genetically tagged CD63 exosomes from the pancreas and observed other organs and cells that were positive for the reporter proteins fused with CD63. Therefore, our data demonstrates that CD63-XFP exosomes were released by pancreas cells and received by other cells.

Regarding phenotypic and functional alterations, as CAFs and endothelial cells were the primary recipients of CD63-XFP exosomes, we investigated potential changes in these cells and observed the following: 1) an inverse correlation between the spatial distribution of aSMA CAFs and Rab27a expression in PDAC lesions; 2) alterations in angiogenesis in healthy pancreas and PDAC in Rab27a KO mice (Fig. 2 and 3).

We have also conducted additional experiments to further address this point: 1) characterization of the protein (mass spectrometry) and RNA (RNA Seq) cargo of small EVs isolated from the pancreas of WT and KPC mice, as well as from a cell line established from a KPC tumor (Fig. 7a-h); 2) exposure of CAFs and endothelial cells to exosomes isolated from a KPC CD63-phiYFP primary cancer cell line and characterization of RNA expression alterations by RNASeq (Fig. 7i-m).

Differentially expressed genes in CAFs that received cancer exosomes are associated with pathways related to cell differentiation, upregulation of cell adhesion, downregulation of cell proliferation and programmed cell death (Fig. 7j). Additionally, genes involved in protein metabolism were altered, reflecting metabolic reprogramming known to influence distinct CAFs behaviors, ultimately impacting cancer cell metabolic adaptations and growth capacity (Li, Sun et al. 2021). We also observed pathways related to the regulation of the immune response, potentially linked with the recently described antigen-presenting CAFs subtype, the biological impact of which is not fully elucidated.

Endothelial cells exposed to cancer exosomes showed downregulation of genes involved in cell adhesion and differentiation, resulting in impaired blood vessel development, as indicated by the Gene Ontology analysis (Fig. 7k). Our data illustrates the capacity of cancer exosomes to modulate the behavior of both CAFs and endothelial cells, aligning with our previous *in vivo* observations (Fig. 2 and 3).

The integrated analysis of KPC small EVs' RNA and protein content, along with the observed alterations in CAFs and endothelial cells upon uptake of cancer exosomes, reveals a direct modulatory effect, as illustrated in Figure 7l-n. Notably, this effect is more pronounced in endothelial cells compared to CAFs, with a higher frequency of identified RNAs in both cell types compared to the protein cargo of KPC small EVs (Fig. 7n). These findings, together with our *in vivo* observations, underscore the modulatory capacity of PDAC EVs in both endothelial cells and CAFs.

A key suggestion of the data is that there is uptake and suggested functional/phenotypic impact of pancreatic or PDAC exosomes on stromal cells. Is there a difference in the 'communication' and the resulting phenotypes of stromal cells between pancreatic and PDAC exosomes?

R: Our observations in PDAC show predominant communication with endothelial cells and CAFs. In the healthy pancreas, we primarily observe communication with endothelial cells. Stellate cells in a healthy pancreas are exceptionally rare (4–7%), and their identification remains challenging (Ferdek and Jakubowska 2017). Nevertheless, when comparing the impact of healthy pancreas and PDAC exosomes on endothelial cells, we noted the following: 1) in Rab27a KO mice there was increased angiogenesis observed in both healthy and PDAC pancreas (Fig. 3e-g); 2) proteomic analysis of small EVs from both WT and PDAC revealed enrichment in angiogenesis-related proteins. This observation aligns with the predominant communication with endothelial cells and the phenotypic alterations observed in Rab27a KO mice *in vivo* (Fig. 2 and 3); 3) by characterizing the alterations that occur in endothelial cells following exposure to cancer exosomes through RNA Seq, we found that these changes were consistent with the phenotypic alterations observed *in vivo* when using PDAC GEMMs (Fig. 3). Collectively, our findings demonstrate that cancer exosomes have the capacity to modulate endothelial cells.

Line229-231: "Collectively, this suggests that modulation of angiogenesis could be more

dependent on the cargo of exosomes rather than quantity/frequency of communication.” This conclusion is questionable as no data is provided to substantiate it. Is there a difference in cargo? Is there a difference in quantity or frequency of exosomes being secreted? While a comprehensive, multiomics analysis of PDAC and healthy exosome cargo would be suitable, the very least would be a thorough evaluation of the frequency of exosomes (healthy/PDAC) in blood/lymph, and correlate those with the occurrences of uptake. See also comment below, Fig5.

R: We acknowledge that the previously mentioned sentence lacked substantiation. To address the raised questions, we conducted mass spectrometry and RNA Seq analysis of the cargo of small EVs from both healthy pancreas and PDAC, demonstrating a significant overlap in their content (Fig. 7b). This likely reflects their common origin and shared biological roles, particularly regarding the observed phenotypic alterations in endothelial cells in both healthy and PDAC contexts.

Proteomic analysis revealed an enrichment in angiogenesis-related proteins in small EVs from both healthy and PDAC samples (Fig. 7c). Additionally, we found that 29% of the small EVs cargo in WT and KPC mice was distinct in proteomic analysis, with 24% of the detected proteins being specific to KPC small EVs, and only 5% specific to WT small EVs. This indicates a more diverse protein repertoire in the cancer context (Fig. 7a,e).

We have also addressed the quantities of exosomes in circulation in healthy and PDAC mice. We observed an increase in the number of serum exosomes by NTA in the presence of cancer (Fig. 6d). Furthermore, we quantified the number of exosomes isolated directly from the pancreas of healthy and PDAC mice, showing a significant increase in PDAC (Fig. 6c).

Despite this clear increase in exosomes numbers, flow cytometry data did not reveal significant differences in the rates of communication with endothelial cells in cancer compared to healthy pancreas. These findings suggest that the modulation of endothelial cells might not be primarily influenced by the quantity of exosomes but is more likely driven by their cargo.

Fig3d-g: While this approach is showing the difference between healthy pancreas and PDAC exosomes on angiogenesis, should the PDAC model not also be performed in a healthy pancreas Rab27ako background? Would PDAC exosomes not also be taken up by healthy pancreas and modulate their (angiogenesis altering) phenotype?

R: We apologize for any confusion, and we believe the reviewer may be referring to the PDAC model in its early stages, which could be considered “histologically healthy pancreas” in comparison to later stages of PDAC development. However, it is important to note that even at these early stages, the pancreas contains transformed cells with KRAS^{G12D} and TP53 mutations, which become active between embryonic days E9.5 to E12.5, when Pdx-1 is active (Stanger, Tanaka et al. 2007). To have a PDAC model in a truly healthy pancreas with a Rab27a KO background would require orthotopically injecting PDAC cells with a Rab27a KO into a healthy pancreas. This approach would no longer represent a spontaneous model, which is the primary reason we developed the ExoBow model.

Similar to above, Fig4 is difficult to interpret as it is not shown if there is a difference locally or systemically between healthy pancreas and PDAC exosomes. The increased uptake (not communication) in immune cells could be attributed to a difference in abundance of exosomes.

R: Our data reveals that despite the significant increase in the percentage of immune cells (CD45⁺) from healthy to PDAC (Fig. 4b), this is not accompanied by a corresponding increase in the percentage of immune cells that have received PDAC exosomes (CD45⁺ PDAC Exos CD63⁺) (Fig. 4a). Therefore, it is important to clarify that there are no differences between the healthy pancreas and PDAC in this regard. This lack of difference is likely due to the increase in CD45⁺ cells being accompanied by a simultaneous increase in exosomes in PDAC (Fig. 6c,d). As a result, the rates of communication are maintained at a similar level in both healthy pancreas and PDAC. We have revised the results section to provide clarity (page 9).

Fig4c and 4d: In 4c, about 8% of T cells are CD63XFP positive, ie took up exosomes. How is it possible that in 4d, each to Th and Tc are about 6% positive, in addition to a small Treg percentage? Should Th + Tc + Treg not be t cells from 4c?

R: To clarify the calculations in Fig. 4c and 4d:

- Fig. 4c: the percentage of T cells corresponds to TCRb⁺ cells that are also positive for CD63-XFP PDAC exosomes.

- Fig. 4d: the percentages are divided into specific T cell subsets: Th (TCRb⁺CD4⁺), Tc (TCRb⁺CD4⁻), and Tregs (CD4⁺FOXP3⁺) that are also positive for CD63-XFP PDAC exosomes.

The percentage of T cells that are positive for CD63-XFP PDAC exosomes can be calculated by considering the proportion of each T cell subset within TCRb⁺ cells.

Here's an example of the calculation for one of our experimental mice:

% of TCRb⁺ cells positive for CD63-XFP PDAC exosomes = 3.3%

% of TCRb⁺CD4⁺ cells positive for CD63-XFP PDAC exosomes = 1.4%

% of TCRb⁺CD4⁻ cells positive for CD63-XFP PDAC exosomes = 4.6%

The frequency of CD4⁺ and CD4⁻ populations within TCRb⁺ cells is 0.5 in each case. Therefore, % of TCRb⁺ cells positive for CD63-XFP PDAC exosomes = (1.4%x0.5) + (4.6%x0.5)= 3%.

Tregs analysis is derived from a separate mix (CD4⁺Foxp3⁺, as detailed in the Methods section. It is not included in these calculations. We provide a gating strategy example in the source data file for reference (<https://figshare.com/s/bb940a672a5642a0a2e5>).

Fig5 and above: The authors achieved to generate a traceable, PDAC-specific exosome model. A key question in the cancer exosome field is the relative amount of cancer exosomes in body fluids, for pancreatic cancer maybe best blood. Using their ExoBow

model, could the authors determine the abundance and clonal heterogeneity of cancer/PDAC exosomes in the blood? Or with the CD63-mCherry at least the abundance in the blood? This would be very informative to support data from Fig5, which while interesting in itself, should be strengthened by exploring the 'blood' and/or 'lymph' intermediate exosome abundance.

R: Thank you for the constructive suggestion. To address this point, we have conducted further experiments:

- 1. We have isolated exosomes from the blood of PDAC CD63-XFP mice at both early and late PDAC stages. Image stream analysis of the beads-exosomes complexes revealed a higher percentage of CD63-XFP exosomes in circulation in mice with an advanced disease stages (Fig. 6e). This finding indicates that the abundance of cancer exosomes increases as the disease progresses, consistent with our observations of greater CD63⁺ exosomes accumulation in distant organs in later disease stages (Fig. 5d).**

Regarding the studies concerning the tumor clonal representation in PDAC exosomes found in the blood, we encountered a technical challenge since a bead-exosomes complex can have different CD63-XFP positive vesicles attached, each with distinct colors. This makes it difficult to quantify the different clonal CD63-XFP populations in the blood accurately. While the suggestion of evaluating lymph in addition to the blood is compelling, practical limitations arise due to the small amounts of fluid collected *in vivo*. This volume constraint hinders the isolation and evaluation of exosomes from lymph. Nonetheless, we have successfully addressed the presence of PDAC exosomes in the blood of mice and have concluded that their abundance increases with disease burden.

SFig7c,d: as above, is this observation a result of less uptake in the organs for healthy pancreas, or a result of reduced healthy pancreas exosome in circulation? Could the above suggested evaluation of blood circulating PDAC exosomes be expanded for healthy pancreas exosomes? The conclusion currently that the increased uptake is due to unexplored content differences is rather far fetched.

R: We conducted new experiments to investigate the abundance of local and systemic EVs in healthy and PDAC conditions. Our findings reveal an enrichment of EVs in PDAC both locally and systemically when compared to healthy state (Fig. 6c,d). This increased EVs presence in PDAC may contribute to the heightened systemic levels of communication observed in PDAC.

Furthermore, we characterized the content of healthy and PDAC small EVs and demonstrated that, while they share a significant overlap in content, PDAC small EVs have a greater number of exclusively identified proteins compared to healthy small EVs (Fig. 7b). Therefore, we cannot exclude the possibility that one or both phenomena contribute to the increased and distinct systemic communication observed in PDAC compared to healthy pancreas. It is likely that not only the quantity and cargo but also the nature of the recipient cells play a role in the observed differences. This topic has been included in our discussion section for further exploration (page 15).

The discussion should be revisited. In general, a restatement of the achievements of the paper should be avoided. The speculation of line340-349 needs to be made more

precise. The suggestion of antigen presentation (line358) is not at all addressed in the study and the rather rudimentary immunological data does not allow any such suggestion. The discussion could focus on applications of the ExoBow model beyond PDAC/pancreatic cancer or cancer in general. What other applications could be found for such a model? What is the current state of 'tracing exosomes' and which problems can be overcome with the ExoBow model? A key limitation is the linkage to CD63 as THE exosome marker. However, several previous publications, including PMID: 26858453 (your reference 13), 34282141, 30949309, show that 'exosomes' are not all CD63 positive.

R: We appreciate your constructive comments, which have significantly enhanced the clarity of our manuscript. Here are the specific changes we have made:

- 1) **We have rephrased and simplified our observations concerning lines 340-349.**
- 2) **We removed the suggested ability of pancreas exosomes to modulate the immune response.**
- 3) **We extended our discussion to address the advantages and potential applications of the ExoBow model.**
- 4) **We highlighted the current state-of-the-art knowledge regarding CD63 expression in different EV subpopulations and emphasized that CD63 is not an absolute marker of every EV of endosomal origin.**

Furthermore, an honest reflection and distinction from other fluorophore or luminescence tracing approaches (for example PMID: 33042758) should be discussed and evaluated.

R: Done. The value of the development of new GEMMs using different reporter systems besides fluorescent ones was added to discussion (page 16).

Minor point:

Please correct the figure legend to 4 from NK to NK1.1

R: Done.

References

- Addi, C., A. Presle, S. Frémont, F. Cuvelier, M. Rocancourt, F. Milin, S. Schmutz, J. Chamot-Rooke, T. Douché, M. Duchateau, Q. Giai Gianetto, A. Salles, H. Ménager, M. Matondo, P. Zimmermann, N. Gupta-Rossi and A. Echard (2020). "The Flemmingsome reveals an ESCRT-to-membrane coupling via ALIX/syntenin/syndecan-4 required for completion of cytokinesis." Nature Communications **11**(1): 1941.
- Baietti, M. F., Z. Zhang, E. Mortier, A. Melchior, G. Degeest, A. Geeraerts, Y. Ivarsson, F. Depoortere, C. Coomans, E. Vermeiren, P. Zimmermann and G. David (2012). "Syndecan-syntenin-ALIX regulates the biogenesis of exosomes." Nature Cell Biology **14**(7): 677-685.
- Carolina, F. R., B. Nuno, A. Barbara, B. Ines, D. Cecilia, A. M. Carlos, A. C. Stephanie, C. L. Francisco, M.-R. Pedro, M. Barbara, C.-P. Ana, S. Soraia, O. Hugo, C. Sergio, C. Jose Luis, G. David, C. Bruno, P. Luisa, K. Tony, M. Guilherme, M. Rui, C. Fatima, C. Marília, K. Raghu, M. Jose Carlos and A. M. Sonia (2022). "Extracellular Vesicles from Pancreatic Cancer Stem Cells Lead an Intratumor Communication Network (EVNet) to fuel tumour progression." Gut **71**(10): 2043.
- Dou, Y., Y. Lin, T. Y. Wang, X. Y. Wang, Y. L. Jia and C. P. Zhao (2021). "The CAG promoter maintains high-level transgene expression in HEK293 cells." FEBS Open Bio **11**(1): 95-104.
- Ferdeck, P. E. and M. A. Jakubowska (2017). "Biology of pancreatic stellate cells-more than just pancreatic cancer." Pflugers Arch **469**(9): 1039-1050.
- Kowal, J., G. Arras, M. Colombo, M. Jouve, J. P. Morath, B. Primdal-Bengtson, F. Dingli, D. Loew, M. Tkach and C. Théry (2016). "Proteomic comparison defines novel markers to characterize heterogeneous populations of extracellular vesicle subtypes." Proceedings of the National Academy of Sciences **113**(8): E968.
- Langemeyer, L., F. Fröhlich and C. Ungermann (2018). "Rab GTPase Function in Endosome and Lysosome Biogenesis." Trends in Cell Biology **28**(11): 957-970.
- Li, W., J. Wang, X. Yin, H. Shi, B. Sun, M. Ji, H. Song, J. Liu, Y. Dou, C. Xu, X. Jiang, J. Li, L. Li, C. Y. Zhang and Y. Zhang (2022). "Construction of a mouse model that can be used for tissue-specific EV screening and tracing in vivo." Front Cell Dev Biol **10**: 1015841.
- Li, Z., C. Sun and Z. Qin (2021). "Metabolic reprogramming of cancer-associated fibroblasts and its effect on cancer cell reprogramming." Theranostics **11**(17): 8322-8336.
- Mathieu, M., N. Névo, M. Jouve, J. I. Valenzuela, M. Maurin, F. J. Verweij, R. Palmulli, D. Lankar, F. Dingli, D. Loew, E. Rubinstein, G. Boncompain, F. Perez and C. Théry (2021). "Specificities of exosome versus small ectosome secretion revealed by live intracellular tracking of CD63 and CD9." Nat Commun **12**(1): 4389.
- McCann, J. V., S. R. Bischoff, Y. Zhang, D. O. Cowley, V. Sanchez-Gonzalez, G. D. Daaboul and A. C. Dudley (2020). "Reporter mice for isolating and auditing cell type-specific extracellular vesicles in vivo." Genesis (New York, N.Y. : 2000) **58**(7): e23369-e23369.
- Men, Y., J. Yelick, S. Jin, Y. Tian, M. S. R. Chiang, H. Higashimori, E. Brown, R. Jarvis and Y. Yang (2019). "Exosome reporter mice reveal the involvement of exosomes in mediating neuron to astroglia communication in the CNS." Nature Communications **10**(1): 4136.
- Neckles, V. N., M. C. Morton, J. C. Holmberg, A. M. Sokolov, T. Nottoli, D. Liu and D. M. Feliciano (2019). "A transgenic inducible GFP extracellular-vesicle reporter (TIGER) mouse illuminates neonatal cortical astrocytes as a source of immunomodulatory extracellular vesicles." Scientific Reports **9**(1): 3094.
- Nørgård, M. Ø., L. B. Steffensen, D. R. Hansen, E.-M. Füchtbauer, M. B. Englund, H. Dimke, D. C. Andersen and P. Svenningsen (2022). "A new transgene mouse model using an extravesicular EGFP tag enables affinity isolation of cell-specific extracellular vesicles." Scientific Reports **12**(1): 496.
- Stanger, B. Z., A. J. Tanaka and D. A. Melton (2007). "Organ size is limited by the number of embryonic progenitor cells in the pancreas but not the liver." Nature **445**(7130): 886-891.
- van Niel, G., G. D'Angelo and G. Raposo (2018). "Shedding light on the cell biology of extracellular vesicles." Nature Reviews Molecular Cell Biology **19**(4): 213-228.

Xie, Y., M. Wang, L. Gu and Y. Wang (2022). "CRISPR/Cas9-mediated knock-in strategy at the Rosa26 locus in cattle fetal fibroblasts." PLoS One **17**(11): e0276811.

Yoshimura, A., M. Kawamata, Y. Yoshioka, T. Katsuda, H. Kikuchi, Y. Nagai, N. Adachi, T. Numakawa, H. Kunugi, T. Ochiya and Y. Tamai (2016). "Generation of a novel transgenic rat model for tracing extracellular vesicles in body fluids." Sci Rep **6**: 31172.

Zhang, H., D. Freitas, H. S. Kim, K. Fabijanic, Z. Li, H. Chen, M. T. Mark, H. Molina, A. B. Martin, L. Bojmar, J. Fang, S. Rampersaud, A. Hoshino, I. Matei, C. M. Kenific, M. Nakajima, A. P. Mutvei, P. Sansone, W. Buehring, H. Wang, J. P. Jimenez, L. Cohen-Gould, N. Paknejad, M. Brendel, K. Manova-Todorova, A. Magalhães, J. A. Ferreira, H. Osório, A. M. Silva, A. Massey, J. R. Cubillos-Ruiz, G. Galletti, P. Giannakakou, A. M. Cuervo, J. Blenis, R. Schwartz, M. S. Brady, H. Peinado, J. Bromberg, H. Matsui, C. A. Reis and D. Lyden (2018). "Identification of distinct nanoparticles and subsets of extracellular vesicles by asymmetric flow field-flow fractionation." Nat Cell Biol **20**(3): 332-343.

REVIEWERS' COMMENTS

Reviewer #1 (Remarks to the Author):

I appreciate the effort the authors spent on addressing the comments. The manuscript has been improved.

Reviewer #2 (Remarks to the Author):

My concerns and edits have been addressed

Reviewer #3 (Remarks to the Author):

The authors of the manuscript Adem et al. have made a tremendous effort to improve the quality and rigor of the manuscript. The addition of a multitude of additional data as well as a sharpening of the text certainly strengthens their findings.

Most of the points I raised in my initial review have been addressed.

In brief, the use of the 'exosome' and 'communication' label have been argued for well, and the publication of the response to the comments by the authors will explain the reasoning to use those in the text to the reader. I support the authors' interpretation of the two labels and request no changes.

As for the additional data as well as the clarification: the authors have answered all queries and concerns I had in sufficient detail. The new data, especially the new Fig7, will be a key resource for the field. Similarly, the improved presentation and discussion of the ExoBow model has been sharpened and should entice a broad interest in it.

Two minor points: I do not support the deposition of the data on a website (figshare), in fact, I abstained from evaluating this data. This data, for example whole the gating strategy and raw data has to be included as supplementary figure.

The proteomic and RNAseq data is described to be 'deposited (...) before publication', which is important to happen in a timely manner to provide the field this important information.

Reviewer #1 (Remarks to the Author):

I appreciate the effort the authors spent on addressing the comments. The manuscript has been improved.

We appreciate the Reviewer positive feedback and thank once again for the valuable input which has significantly enhanced the quality of our manuscript.

Reviewer #2 (Remarks to the Author):

My concerns and edits have been addressed

We appreciate the Reviewer positive feedback and thank for the overall constructive comments which helped us improve our manuscript.

Reviewer #3 (Remarks to the Author):

The authors of the manuscript Adem et al. have made a tremendous effort to improve the quality and rigor of the manuscript. The addition of a multitude of additional data as well as a sharpening of the text certainly strengthens their findings.

Most of the points I raised in my initial review have been addressed.

In brief, the use of the 'exosome' and 'communication label' have been argued for well, and the publication of the response to the comments by the authors will explain the reasoning to use those in the text to the reader. I support the authors' interpretation of the two labels and request no changes.

As for the additional data as well as the clarification: the authors have answered all queries and concerns I had in sufficient detail. The new data, especially the new Fig7, will be a key resource for the field. Similarly, the improved presentation and discussion of the ExoBow model has been sharpened and should entice a broad interest in it.

We thank the Reviewer for the positive feedback and the overall dedication in the peer-review process which has greatly contributed for the improvement of our work.

Two minor points: I do not support the deposition of the data on a website (figshare), in fact, I abstained from evaluating this data. This data, for example whole the gating strategy and raw data has to be included as supplementary figure.

We have now included the flow cytometry gating strategy in Supplementary Figure 9b and 10. Due to space constraints, we have further detailed this information in the Source Data file, which will be published together with the manuscript. Thus, all supplementary and raw data will be readily accessible.

The proteomic and RNAseq data is described to be deposited (...) before publication', which is important to happen in a timely manner to provide the field this important information.

Proteomics and RNA Seq data are now fully available in PRIDE and European Nucleotide Archive (ENA) browser, respectively. The accession codes and hyperlinks are specified in the manuscript section "Data availability". Proteomics accession code: PXD047009

(<https://www.ebi.ac.uk/pride/archive/projects/PXD047009>). RNA Seq accession code:
PRJEB71061 (<https://www.ebi.ac.uk/ena/browser/view/PRJEB71061>).